# MuCO: Generative Peptide Cyclization Empowered by Multi-stage Conformation Optimization

Yitian Wang [1 2]   Fanmeng Wang [1 2]   Angxiao Yue [1]   Wentao Guo [2]   Yaning Cui [2]   Hongteng Xu [1 3 4]

## Abstract

Modeling peptide cyclization is critical for the virtual screening of candidate peptides with desirable physical and pharmaceutical properties. This task is challenging because a cyclic peptide often exhibits diverse, ring-shaped conformations, which cannot be well captured by deterministic prediction models derived from linear peptide folding. In this study, we propose MuCO (Multi-stage Conformation Optimization), a generative peptide cyclization method that models the distribution of cyclic peptide conformations conditioned on the corresponding linear peptide. In principle, MuCO decouples the peptide cyclization task into three stages: topology-aware backbone design, generative side-chain packing, and physics-aware all-atom optimization, thereby generating and optimizing conformations of cyclic peptides in a coarse-to-fine manner. This multi-stage framework enables an efficient parallel sampling strategy for conformation generation and allows for rapid exploration of diverse, low-energy conformations. Experiments on the large-scale CPSea dataset demonstrate that MuCO significantly and consistently outperforms state-of-the-art methods in physical stability, structural diversity, secondary structure recovery, and computational efficiency, making it a promising computational tool for exploring and designing cyclic peptides. The demo of the proposed method can be found at https://github.com/mianqiu00/MuCO.

## 1. Introduction

Cyclic peptides constitute a significant subset for drug discovery, chemical biology, and material design (Jing & Jin, 2020; Ji et al., 2024; Song et al., 2021), whose ring-closed conformations provide them with enhanced stability, proteolytic resistance, membrane permeability, and target-binding specificity compared to the corresponding linear peptides (Dougherty et al., 2019; Mannes et al., 2022; Lee et al., 2019). In general, a cyclic peptide exists as highly dynamic conformational ensembles exploring a complex energy landscape shaped by geometric constraints (Schlick et al., 2021; Jiang et al., 2025; Zorzi et al., 2017). *Peptide cyclization* refers to predicting cyclic peptide conformations given linear peptide sequences or conformations. It helps identify unobserved potential wells and explore hidden conformations governing biological activity (Persch et al., 2015; Sekhar & Kay, 2013), which is essential for the virtual screening of peptides with desirable physical and pharmaceutical properties.

Currently, physical simulation methods (Pracht et al., 2020; Hollingsworth & Dror, 2018) offer relatively high precision in peptide cyclization. However, they are computationally prohibitive for large-scale exploration, which motivates the development of learning-based methods. Focusing on peptide cyclization, the general protein folding method, such as AlphaFold3 (AF3) (Abramson et al., 2024), remains suboptimal, struggling with ring closure and atomic clashes (Zhang et al., 2025). Specialized methods like HighFold2 (Zhu et al., 2025) and AfCycDesign (Rettie et al., 2025) are based on AlphaFold2 (AF2) (Jumper et al., 2021) and cyclize peptides by deterministic folding, thereby failing to generate diverse conformations. As a result, given a linear peptide, how to generate cyclic peptide conformations with guaranteed physical rationale and diversity is still an open problem.

In this study, we propose **MuCO**, an effective conformation generation method for cyclic peptides, which achieves promising peptide cyclization via **Mu**lti-stage **C**onformation **O**ptimization. As illustrated in Figure 1, MuCO decouples peptide cyclization into three stages, including $i$) *topology-aware cyclic backbone generation*, $ii$) *generative side-chain packing*, and $iii$) *physics-aware conformation optimization*. Given a linear peptide, the first stage generates diverse back-

[1]Gaoling School of Artificial Intelligence, Renmin University of China, Beijing, China [2]DP Technology, Beijing, China [3]Beijing Key Laboratory of Research on Large Models and Intelligent Governance [4]Engineering Research Center of Next-Generation Intelligent Search and Recommendation. Correspondence to: Hongteng Xu <hongtengxu@ruc.edu.cn>.

*Proceedings of the 43rd International Conference on Machine Learning*, Seoul, South Korea. PMLR 306, 2026. Copyright 2026 by the author(s).

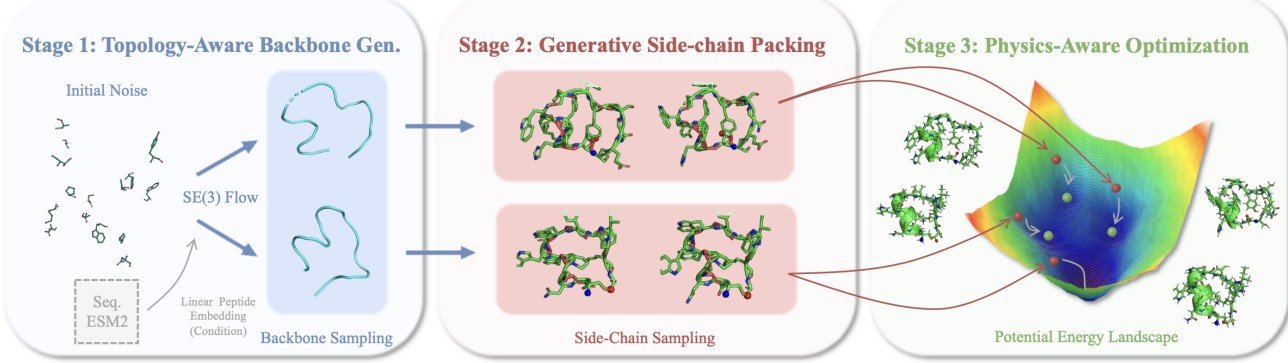

*Figure 1.* An illustration of the 3-stage scheme of MuCO. The generation process is decoupled into three stages, including $i$) **Topology-Aware Backbone Generation** for backbone scaffold $B$ by SE(3) flow matching on condition of sequence embedding, $ii$) **Generative Side-chain Packing** for side chain sampling via conditional flow matching with cyclic RPE, $iii$) **Physics-Aware Optimization** by CHARMM36 forcefield to yield final energy minimized conformation.

bone scaffolds that satisfy the ring-closure (Huguet et al., 2024). The second stage finds optimal rotamer configurations for each backbone, generating full-atom conformations through side-chain sampling and suppressing the risk of steric clashes (Lee & Kim, 2025). The third stage applies CHARMM36 (Huang & MacKerell Jr, 2013) to fine-tune the local geometry of each conformation under potential energy minimization, guiding the conformations into a valid and stable molecular space.

Compared to existing methods (Zhu et al., 2025; Zhang et al., 2025), MuCO has advantages on both computational efficiency and generation quality. In particular, unlike the end-to-end strategy that attempts to simultaneously predict or generate all atoms, the multi-stage framework of MuCO supports an efficient parallel sampling strategy that rapidly identifies potential wells and various cyclization modes that are often inaccessible in single-shot or several-shot predictors. As a result, MuCO can generate reliable conformations with high efficiency. Experiments on the large-scale CPSea dataset (Yang et al., 2026) demonstrate that MuCO significantly outperforms state-of-the-art methods (Zhu et al., 2025), particularly in physical stability, structural diversity, secondary structure recovery, and computational efficiency.

## 2. Related Work

### 2.1. Peptide Folding and Design

The prediction of peptide conformation has largely followed the protein folding paradigm. AF2 (Jumper et al., 2021) accurately folds peptides using MSAs, while ESMFold (Lin et al., 2022) leverages protein language models (PLMs) to infer structures directly from single sequences, bypassing the need for alignment searching. Recently, generative modeling has expanded this frontier: AlphaFold3 (Abramson et al., 2024), Chai-1 (team et al., 2024), Boltz-1 (Wohlwend et al., 2025; Passaro et al., 2025), and SimpleFold (Wang et al., 2025b) utilize diffusion-based or iterative refinement

modules to generate high-fidelity all-atom conformations for peptides.

Unlike the above end-to-end folding methods, some methods generate peptide conformations by two steps: backbone design and side-chain packing. For backbone design, FrameDiff (Yim et al., 2023b) and FrameFlow (Yim et al., 2023a) apply SE(3) diffusion and flow matching to generate protein backbones, respectively. Based on them, FoldFlow (Bose et al., 2024) and FoldFlow2 (Huguet et al., 2024) further integrate optimal transport and sequence conditioning (via ESM based embeddings (Lin et al., 2022)) to generate diverse scaffold topologies. ReQFlow (Yue et al., 2025) improves the generation efficiency using rectified quaternion flows. For side-chain packing, methods like Diff-Pack (Zhang et al., 2023), H-Packer (Visani et al., 2024), and FlowPacker (Lee & Kim, 2025) treat rotamer packing as a generative problem, learning the conditional distribution of side-chains given a fixed backbone. These decoupled generation methods have been widely used in protein design, which is applicable for generating peptide conformations.

The above methods achieve a promising progress in linear peptide folding and design. However, they are often inapplicable in cyclic peptide generation due to a lack of appropriate ring-closure constraints.

### 2.2. Cyclic Peptide Modeling and Generation

Focusing on cyclic peptide modeling and generation, the mainstream solution is adapting models pretrained on linear peptides. AfCycDesign (Rettie et al., 2025) and High-Fold2 (Zhu et al., 2025) modify the Relative Positional Encoding (RPE) of AF2 to ensure ring-closure. CP-Composer (Jiang et al., 2025) decomposes cyclic constraints into linear sub-units and achieves zero-shot cyclic peptide design. These methods, however, are not trained on true cyclic peptides because of the data scarcity issue, which limits their abilities to find ring-closed, low-energy confor-

mations. As a result, they often fail to model the specific strain and distortions introduced by cyclization.

The recent release of the CPSea dataset (Yang et al., 2026) provides the first large-scale resource of cyclic peptide conformations, enabling a paradigm shift from linear-to-cyclic adaptation to native cyclic generation. By leveraging the dataset, our work proposes a multi-stage conformation optimization strategy for generative peptide cyclization, generating high-quality conformational ensembles of cyclic peptides efficiently.

## 3. Proposed Method

### 3.1. Problem Formulation and Modeling Principle

As mentioned above, we formulate peptide cyclization as a conditional generation problem. Denote $\mathcal{S} = \{s_1, \ldots, s_L\}$ as a linear peptide of length $L$, where $s_i$ represents the type of the $i$-th residue. Denote $\boldsymbol{X} \in \mathbb{R}^{N \times 3}$ as the $N$ 3D atomic coordinates of the corresponding cyclic peptide. Following the rigid-frame convention (Jumper et al., 2021), we can parametrize $\boldsymbol{X}$ equivalently by the rigid transformation of backbone and internal torsion angles of side chains:

- **Backbone scaffold $\mathcal{B}$:** The backbone geometry is represented by the tuple $\mathcal{B} = \{(\boldsymbol{T}_i, \psi_i)\}_{i=1}^{L}$. Here, $\boldsymbol{T}_i = (\boldsymbol{R}_i, \boldsymbol{t}_i) \in \mathrm{SE}(3)$ denotes the rigid transformation of the local frame of the $i$-th residue (constructed from the "$\mathrm{N} - \mathrm{C}_\alpha - \mathrm{C} - \mathrm{O}$" structure of residue), where $\boldsymbol{R}_i \in \mathrm{SO}(3)$ is the rotation matrix and $\boldsymbol{t}_i \in \mathbb{R}^3$ is the translation vector. Additionally, $\psi_i \in \mathbb{T}$ is the backbone torsion angle determining the position of the carbonyl oxygen atom, where $\mathbb{T} = \mathbb{R}/2\pi\mathbb{Z}$ denotes the 1-torus representing the angular periodicity.

- **Side chains $\mathcal{C}$:** The conformation of side-chains is parameterized by a set of torsion angles $\mathcal{C} = \{\boldsymbol{\chi}_i\}_{i=1}^{L}$. For each residue $s_i$, the side chain is determined by up to four angles $\boldsymbol{\chi}_i = [\chi_{i,1}, \ldots, \chi_{i,4}] \in \mathbb{T}^4$.

This geometric data representation captures rotational invariance and local rigidity of conformation, and the atomic coordinates $\boldsymbol{X}$ can be recovered by the Forward Kinematics function, i.e., $\boldsymbol{X} = \mathrm{FK}(\mathcal{B}, \mathcal{C})$.

We aim to learn the conditional distribution $p(\boldsymbol{X}|\mathcal{S})$ and sample diverse conformations that are ring-closure and low-energy efficiently from $p(\boldsymbol{X}|\mathcal{S})$, which leads to the proposed MuCO method.

In principle, MuCO leverages the equivalence between $p(\boldsymbol{X}|\mathcal{S})$ and $p(\mathcal{B}, \mathcal{C}|\mathcal{S})$ and the hierarchical structure of $p(\mathcal{B}, \mathcal{C}|\mathcal{S})$ to model $p(\boldsymbol{X}|\mathcal{S})$ in a decomposed format:

$$p(\boldsymbol{X}|\mathcal{S}) = p(\mathcal{B}, \mathcal{C}|\mathcal{S}) = \underbrace{p(\mathcal{B}|\mathcal{S})}_{\text{Backbone}} \cdot \underbrace{p(\mathcal{C}|\mathcal{B}, \mathcal{S})}_{\text{Side chains}}. \quad (1)$$

Here, $p(\mathcal{B}|\mathcal{S})$ is the conditional distribution of the backbone scaffold $\mathcal{B}$ given the residue sequence $\mathcal{S}$, and $p(\mathcal{C}|\mathcal{B}, \mathcal{S})$ is the conditional distribution of the side chains $\mathcal{C}$ given $\mathcal{B}$ and $\mathcal{S}$. Based on this decomposition, MuCO achieves a three-stage conformation optimization pipeline. As illustrated in Figure 1, it generates multiple cyclic peptide conformations given a linear peptide $\mathcal{S}$ by

1) Backbone generation: $\{\mathcal{B}_k\}_{k=1}^{K} \sim p_\theta(\cdot|\mathcal{S})$,

2) Side-chain packing: $\{\mathcal{C}_m^k\}_{m=1}^{M} \sim p_\tau(\cdot|\mathcal{B}_k, \mathcal{S}), \; \forall k$,

3) Conformation refinement: $\widetilde{\boldsymbol{X}}_m^k = \mathrm{FK}(\mathcal{B}_k, \mathcal{C}_m^k)$,

$$\boldsymbol{X}_m^k = \arg\min_{\boldsymbol{X}} E(\boldsymbol{X}) + R(\boldsymbol{X}, \widetilde{\boldsymbol{X}}_m^k), \; \forall k, m. \quad (2)$$

The first stage samples $K$ multiple cyclic backbones, the second stage generates $K \times M$ conformations by sampling $M$ side-chain rotamers for each backbone. Finally, the third stage achieves energy-driven conformation refinement, where $E$ denotes the energy functional of atoms and $R$ denotes the regularization term determined by the initial conformation $\widetilde{\boldsymbol{X}}_m^k$. Here, $p(\mathcal{B}|\mathcal{S})$ and $p(\mathcal{C}|\mathcal{B}, \mathcal{S})$ are parameterized by two generative models, whose parameters are denoted as $\theta$ and $\tau$, respectively. In such a situation, the first two stages can be executed in parallel on GPUs, which reduces the amortized runtime per conformation significantly.

*The key point of MuCO lies in ensuring the above three stages achieve consistent conformation optimization.* In particular, Stage-1 should generate backbones with guaranteed ring-closure structures. Stage-2 should resolve steric clashes in dense environments. Stage-3 should refine the all-atom conformations, maintaining the ring-closure structures and avoiding steric clashes. Moreover, the second and third stages should reduce the potential energy of conformation consistently. In the following content, we will introduce the technical details of MuCO and show how the above requirements are satisfied.

### 3.2. Multi-stage Conformation Optimization

Figure 2 illustrates the computational pipeline of MuCO. To meet the above requirements, our MuCO method imposes implicit cyclization constraints into the model by training on the cyclic peptide dataset and applies force field optimization in Stage-3, enabling cyclization to emerge naturally with guarantees in rationality and diversity. Detailed training strategy is provided in Appendix A.

**Stage-1: Topology-Aware Cyclic Backbone Generation.** Following FoldFlow2 (Huguet et al., 2024), we employ a sequence-conditioned SE(3) flow matching framework for cyclic backbone generation. We model a conditional probability path $p_t(\mathcal{B}|\mathcal{S})$ that interpolates between a simple prior distribution $p_0$ (typically a Gaussian distribution on $\mathbb{R}^3$ and isotropic distribution on SO(3)) and the data distribution $p_1$ over time $t \in [0, 1]$. This process is governed by

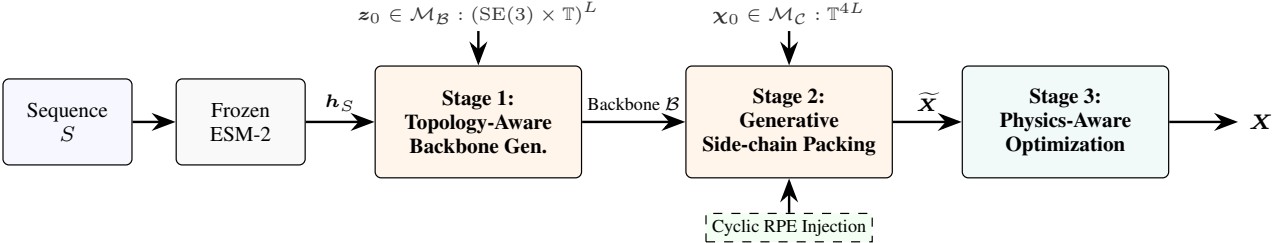

*Figure 2.* MuCO framework overview. The process decouples into three stages: (1) cyclic backbone generation on $\mathcal{M}_\mathcal{B}$, (2) side-chain packing on $\mathcal{M}_\mathcal{C}$ with Cyclic RPE, and (3) physics-aware refinement to reach the local energy minimum $\boldsymbol{X}$.

the velocity field on the manifold of backbone, denoted as $\mathcal{M}_\mathcal{B} := \mathrm{SE}(3)^\mathrm{L} \times \mathbb{T}^\mathrm{L}$. We parameterize the velocity field $v_\theta^\mathcal{B}(\boldsymbol{z}_t, t, \boldsymbol{h}_S)$ by an Invariant Point Attention (IPA) Transformer (Jumper et al., 2021), where $t \in [0, 1]$, $\boldsymbol{h}_S$ is the embedding of the linear peptide, derived from a pre-trained ESM-2 model (Lin et al., 2022). $\boldsymbol{z}_t$ is the backbone state at time $t$, where $\boldsymbol{z}_0 = \{\boldsymbol{T}_i^0 = (\boldsymbol{R}_i^0, \boldsymbol{t}_i^0)\}_{i=1}^L$ is noise sampled from the prior distribution, $\boldsymbol{z}_1 = \{\boldsymbol{T}_i^1 = (\boldsymbol{R}_i^1, \boldsymbol{t}_i^1)\}_{i=1}^L$ is observed data, and $\boldsymbol{z}_t = \{\boldsymbol{T}_i^t = (\boldsymbol{R}_i^t, \boldsymbol{t}_i^t)\}_{i=1}^L$ with $\boldsymbol{t}_i^t = (1-t) \cdot \boldsymbol{t}_i^0 + t \cdot \boldsymbol{t}_i^1$ and $\boldsymbol{R}_i^t = \exp_{\boldsymbol{R}_i^0}(t \cdot \log_{\boldsymbol{R}_i^0}(\boldsymbol{R}_i^1))$.

Applying Riemannian flow matching (Bose et al., 2024; Chen & Lipman, 2024), we can train the model via

$$\min_\theta \mathbb{E}_{t, \boldsymbol{z}_1, \boldsymbol{z}_t} \left\| v_\theta^\mathcal{B}(\boldsymbol{z}_t, t, \boldsymbol{h}_S) - u_t^\mathcal{B}(\boldsymbol{z}_t | \boldsymbol{z}_1) \right\|_{\mathcal{TM}}^2, \quad (3)$$

where $\|\cdot\|_{\mathcal{TM}}$ denotes the Riemannian metric on the tangent space, $u_t^\mathcal{B}$ is the velocity field determined by $\boldsymbol{z}_t$. For the backbone torsion angles $\{\psi_i\}_{i=1}^L$, we can generate them by passing the last-layer embeddings of $v_\theta^\mathcal{B}$ by an MLP.

**Remark 1: Training on cyclic peptides leads to implicit ring-closure constraints.** Here, a critical challenge is to ensure the ring-closure structures of the generated backbones. Traditional methods train models on linear peptides and impose hard geometric constraints during generation (Rettie et al., 2025; Zhu et al., 2025) or apply post-hoc energy minimization to force closure (Zhang et al., 2025), which may introduce unnatural physical bond strains or atom clashes. With the help of the CPSea dataset (Yang et al., 2026), we can train our model directly on cyclic peptides (see the data construction part of the following experimental section), making cyclization an intrinsic geometric property encoded within the conformational subspace. After training on cyclic peptides, the ring-closure structures are fitted by the learned velocity field $v_\theta^\mathcal{B}$ — the model captures the global trajectory of all residues, whose endpoints are cyclic backbones with high probabilities. Consequently, the generated backbones serve as geometrically valid scaffolds for the subsequent packing stage.

**Stage-2: Conditional Generative Side-chain Packing.** Once the backbone scaffold $\mathcal{B}$ is determined, the next challenge is to fill it with side-chain rotamers $\mathcal{C}$. We explicitly model the conditional distribution $p_\tau(\mathcal{C}|\mathcal{B}, \mathcal{S})$ in the Torsional Flow Matching (TFM) framework (Lee & Kim, 2025).

In particular, the space of side chain is a high-dimensional hypertorus $\mathcal{M}_\mathcal{C} = \mathbb{T}^{4L}$, where each dimension corresponds to a periodic torsion angle $\chi$. We learn a velocity field $v_t^\mathcal{C} : \mathbb{T}^{4L} \times [0, 1] \times \mathrm{SE}(3)^L \to \mathcal{T} \times \mathbb{T}^{4L}$ that transports noise sampled from a uniform prior distribution from the torus sampled from the target rotamer distribution. The optimization problem of TFM is

$$\min_\tau \mathbb{E}_{t, \boldsymbol{\chi}_1, \boldsymbol{\chi}_t} \left\| v_\tau^\mathcal{C}(\boldsymbol{\chi}_t, t, \mathcal{B}, \mathcal{S}) - u_t^\mathcal{C}(\boldsymbol{\chi}_t | \boldsymbol{\chi}_1) \right\|_\mathbb{T}^2, \quad (4)$$

where $v_\tau^\mathcal{C}$ is parameterized by Equivariant Graph Neural Network (EGNN), $u_t^\mathcal{C}$ is the conditional vector field derived from the geodesic path on the torus, and $\boldsymbol{\chi}_t$ is the interpolation between the random noise $\boldsymbol{\chi}_0$ and sampled data $\boldsymbol{\chi}_1$ at time $t$. Notably, $v_\tau^\mathcal{C}$ is conditioned on the backbone $\mathcal{B}$ and the sequence $\mathcal{S}$. We extract E(3)-invariant geometric features from $\mathcal{B}$ and fuse them with sequence embeddings $h_\mathcal{S}$ as the input of $v_\tau^\mathcal{C}$. Crucially, by modeling the *distribution* rather than regressing a single conformation (Wang et al., 2025a), we gain the flexibility to sample multiple plausible rotameric states for residues under high steric stress, effectively breathing within the tight rigid cyclic backbone.

**Remark 2: Cyclic-aware graph encoding mitigates clashes.** Existing packing models like DiffPack (Zhang et al., 2023) and FlowPacker (Lee & Kim, 2025) rely on linear relative positional encodings (RPE), which assume that residue 1 and residue $L$ are spatially and sequentially distant. This assumption breaks down in cyclic peptides: The geometric constraints of ring-closure often force side chains to be folded into a compact space, creating a high-density environment where packers struggle to find clash-free solutions. Similar to existing methods (Rettie et al., 2025; Zhu et al., 2025), we construct a residue graph $\mathcal{G}_{\mathrm{cyc}}$ for each $\mathcal{B}$ and augment $v_\tau^\mathcal{C}$ with a cyclic-aware graph encoding mechanism to resolve the clashing issue. In particular, denote the adjacency matrix of $\mathcal{G}_{\mathrm{cyc}}$ as $\boldsymbol{A} = [a_{ij}] \in \{0, 1\}^{L \times L}$, $a_{ij} = 1$ iff the residues $s_i$ and $s_j$ are adjacent. Accordingly, for any two residues $i$ and $j$, the cyclic topological distance, denoted as $d_{\mathrm{cyc}}(i, j)$, is defined as the shortest path distance between them. Given this cyclic metric, we replace the linear RPE with a **Cyclic Relative Positional Encoding**: injecting the metric into the node-to-node attention bias of the underlying EGNN (e.g., EquiformerV2 (Liao et al., 2024)). By topologically rewiring the graph, the model perceives the

boundary residues as immediate neighbors. This allows our model to generate coordinated side-chain packing across the cyclization junction, suppressing critical steric clashes.

Training details of Stage-1 and Stage-2 are provided in Appendix A.

**Stage-3: Physics-Aware Conformation Optimization.** While the preceding flow matching stages effectively navigate the global conformational landscape to identify favorable high-probability regions, the generated atomic coordinates are inherently stochastic samples from a learned distribution. For each generated conformation, while the global topology might be reasonable, its local atomic placements often exhibit minor deviations from ideal bond lengths or van der Waals radii, leading to high-energy steric clashes. This is particularly pronounced in sampling-based approaches compared to deterministic prediction methods, as the model prioritizes distributional diversity over mean-squared-error minimization. To resolve this issue, we implement a physics-aware refinement stage like CP-Composer (Jiang et al., 2025) does, applying a CHARMM36-based force field engine (Huang & MacKerell Jr, 2013) to optimize generated conformations and filtering out invalid ones.

To automate this process for diverse cyclization modes without manual intervention, we design a rule-based topology detection algorithm based on open-source pipeline provided by CP-Composer (Jiang et al., 2025) and CPSea (Yang et al., 2026), which can automatically infer the intended covalent connectivity based on the proximity of reactive groups in the generated conformation $\widetilde{X}$, and apply our updated force field engine to minimize the energy (detailed algorithms provided in Appendix B).

**Remark 3: The guarantee on energy minimization.** Notably, this process corresponds to the energy-driven conformation optimization shown in (2). When applying CHARMM36, we design the $E(\boldsymbol{X})$ in (2) as the potential energy of $\boldsymbol{X}$ based on atomic coordinates and bonds, and implement the $R(\boldsymbol{X}, \widetilde{\boldsymbol{X}})$ in (2) as $\frac{w}{2}\|\boldsymbol{X} - \widetilde{\boldsymbol{X}}\|_F^2$, with a weight $w > 0$. In such a situation, CHARMM36 optimizes conformation by force-field relaxation with Langevin thermostat (Davidchack et al., 2009), reducing the potential energy monotonically during optimization.

In summary, MuCO leverages the best of both worlds: the generative flow efficiently jumps across high-energy barriers to explore diverse potential wells, while the physics based optimization ensures that the final structures descend into valid, physically realizable local minima.

### 3.3. Efficient Exploration via Hierarchical Sampling

A critical advantage of MuCO's decoupled framework over monolithic end-to-end models (e.g., HighFold2) is the capability for efficient, tree-structured parallel sampling. Since the backbone generation (Stage-1) and side-chain packing (Stage-2) are disentangled, we can employ a hierarchical $K \times M$ sampling strategy that significantly expands the exploration of the conformational landscape with minimal computational overhead.

**Tree-Structured Inference.** Instead of performing expensive full-atom recycling for every sample, we first generate $K$ diverse backbone scaffolds using the lightweight backbone flow model. For generated backbones $\{\mathcal{B}_k\}_{k=1}^K$, we then sample $K \times M$ distinct side-chain packing configurations $\{\mathcal{C}_m^k\}_{k,m=1}^{K,M}$ in parallel using the generative packing model. This approach leverages the reduced parameter size of our specialized sub-modules, enabling high-throughput generation orders of magnitude faster than repeatedly running large-scale AF2-based models (Zhu et al., 2025; Rettie et al., 2025).

**Broad Potential Well Coverage.** This hierarchical strategy is crucial for the success of the subsequent Physics-Aware Optimization (Stage-3). As aforementioned, the energy landscape of cyclic peptides is rugged, containing numerous deep, narrow local minima separated by high energy barriers. By generating a massive ensemble of conformations ($K \times M$), MuCO effectively seeds a wide variety of potential wells. Consequently, the force-field relaxation does not need to traverse high barriers globally but simply descends from these diverse initialized points into their respective local minima. This greatly increases the probability of locating the true global minimum and discovering novel, low-energy metastable states that are often missed by deterministic prediction (Zhu et al., 2025).

## 4. Experiments

We evaluate the performance of MuCO in physical stability, structural diversity, secondary structure recovery, and computational efficiency. In particular, we conduct benchmarks on the curated CPSea dataset to demonstrate the advantages of our MuCO over state-of-the-art methods in peptide cyclization. All inference experiments are conducted on a single NVIDIA RTX 4090 GPU.

The following subsections highlight the most representative experimental results. We provide exhaustive numerical comparisons across all sub-datasets and a wide array of visualizations in the Appendix F.

### 4.1. Experimental Setup

**Dataset Construction.** Besides developing MuCO, another contribution of our work is constructing a large-scale, paired benchmark dataset for peptide cyclization based on the CPSea repository (Yang et al., 2026). In particular, for each target cyclic conformation $\boldsymbol{X}_{\text{cyc}}$ in CPSea, we gener-

ated a corresponding linear precursor $X_{\text{lin}}$ using SimpleFold 3M (Wang et al., 2025b) based on the amino acid sequence $S$. Given all linear-cyclic peptide pairs, we apply a rigorous data filtering pipeline, constructing the proposed dataset using those high-confidence conformation pairs. Finally, this dataset is partitioned into mutually exclusive subsets based on functional annotations to prevent data leakage: **CPSea-Train** (The conformation pairs for training and validation), **CP-Bind** (high affinity), **CP-Trans** (cell-penetrating), **CP-Core** (initial intersection subset of CP-Bind and CP-Trans), and **CPSea-PDB** (gold standard data from Protein Data Bank). The last four subsets lead to a testing set containing 10,132 cyclic conformations in total. These conformations work as the Ground Truth (**GT**) and provide references when evaluating model performance. Detailed filtering strategy and analysis of datasets are provided in Appendix C.

**Baselines and Implementations.** We compare MuCO against two categories of state-of-the-art methods:

$i$) Peptide folding models adapted from AF2, including **HighFold2** (Zhu et al., 2025) and **AfCycDesign** (Rettie et al., 2025). These two models are pretrained on the AF2 dataset that has covered CPSea.

$ii$) Geometric Deep Learning (GDL) models for all-atom conformation prediction and optimization, including **EGNN** (Satorras et al., 2021) and **WGFormer** (Wang et al., 2025a). These two models are trained to predict cyclic conformations based on the linear ones.

Notably, all the baselines apply our physics-aware optimization method (i.e., Stage-3 of MuCO) to guarantee a reasonable success rate when generating cyclic conformations. In addition, when evaluating the model performance in generation quality, MuCO applies the single sampling mode (i.e. setting $K = M = 1$) for a fair comparison. Training details of the baselines are provided in Appendix A.

**Metrics.** We evaluate performance using three key dimensions: $i$) **Success Rate**, defined as the percentage of generated conformations satisfying geometric bond criteria; $ii$) **Physical Stability**, measured by the mean potential energy ($E$) calculated via CHARMM36; and $iii$) **Diversity**, quantified by the Shannon Entropy ($H$) of cyclization modes[1] ($H_{\mathcal{C}}$) and secondary structure clusters ($H_{SS}$). Furthermore, to assess pharmaceutical potential and local structural fidelity, we expand our evaluation to **Physicochemical Properties** (including H-bond counts, radius of gyration $R_g$, and solvent-accessible surface area SASA) and **Jensen-Shannon (JS) Similarity** for backbone/side-chain

---

[1]We consider three cyclization modes in this study, including $i$) **H2T** (Head-to-Tail cyclization mode), $ii$) **C2C** (Cys-Cys mode corresponding to Disulfide), $iii$) **K2DE** (Isopeptide bond of K(Lys)-D(Asp) or K-E(Glu) in side chains). Detailed definitions are provided in Appendix B.1.

torsions and secondary structures. Detailed descriptions of the metrics are provided in Appendix D.

*Table 1.* The average performance on the testing set. In the single sampling mode, MuCO achieves the best balance between physical stability and structural diversity, significantly outperforming baselines in energy minimization while avoiding mode collapse.

| Method | Succ.↑ | Energy↓ | Div.↑ | Mode Distribution (%) | | |
|---|---|---|---|---|---|---|
| | (%) | (kJ/mol) | ($H$) | H2T | K2DE | C2C |
| GT (Ref.) | - | -15.8 | 0.59 | 23.2 | 72.6 | 4.2 |
| EGNN | 85.6 | 108.2 | 0.57 | 22.5 | 75.2 | 2.3 |
| WGFormer | 86.2 | 104.9 | 0.58 | 21.6 | 76.0 | 2.4 |
| AfCycDesign | 98.4 | 116.4 | 0.01 | 99.9 | 0.1 | 0.0 |
| HighFold2 | **98.9** | 111.3 | 0.00 | 99.9 | 0.0 | 0.0 |
| **MuCO (Ours)** | 94.0 | **30.2** | **0.69** | 49.0 | 50.5 | 0.6 |

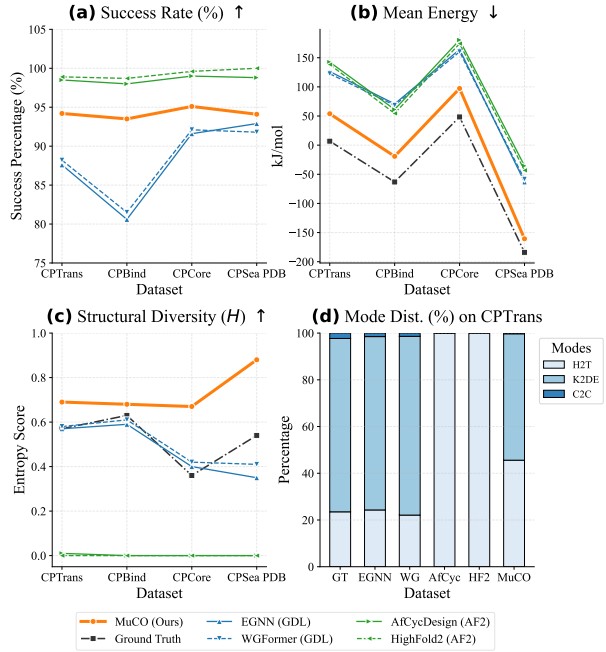

*Figure 3.* (a) Success Rate (%): Percentage of successful cyclization. (b) Mean Potential Energy ($E$): Physical stability calculated via CHARMM36. (c) Structural Diversity ($H_{\mathcal{C}}$): Shannon entropy of conformational clusters. (d) Mode distribution coverage on the CPTrans dataset.

### 4.2. Main Results: Generation Quality and Diversity

**MuCO Overcomes Mode Collapse with Lower Energy.** We conduct baselines and a single sample on MuCO to fairly compare their effectiveness in cyclization. Table 1 and Figure 3 present the quantitative comparison of four test sets. A critical observation is the trade-off between Success Rate and Physical Stability. Although AF2-based methods (HighFold2 and AfCycDesign) achieve near-perfect success rates ($> 98\%$), they suffer from severe mode collapse, showing zero diversity ($H \approx 0$) and failing to cover alternative cyclization modes (e.g., virtually 0% coverage for

K2DE and Cys2Cys modes). Furthermore, their predicted conformations often reside in high-energy states, indicating unresolved steric clashes. In contrast, MuCO achieves the lowest potential energy across all datasets, significantly outperforming the baselines. Crucially, MuCO maintains high structural diversity, successfully recovering a balanced distribution of cyclization modes compared to the ground truth. Notably, MuCO reaches a success rate of around 95% even in the single sampling mode. These results confirm that our method effectively learns a physically meaningful and diverse conformational ensemble rather than memorizing a single mean conformation.

**Secondary Structure Recovery.** Beyond global topology, we further analyze local secondary structures in Figures 4 and 5. EGNN and WGFormer fail to form regular structures, especially in $\beta$-sheets, resulting in excessive random coils ($> 90\%$) with high energy. Although AF2-based models capture more secondary structure information, the structural diversity remains far behind that of the ground truth. MuCO significantly improves the recovery of $\alpha$-helices and $\beta$-sheets, closely matching the distribution of the Ground Truth conformations. The high diversity score ($H = 0.8$ on average) further indicates that MuCO generates a rich mixture of secondary structures, ensuring the functional versatility of the generated cyclic peptides.

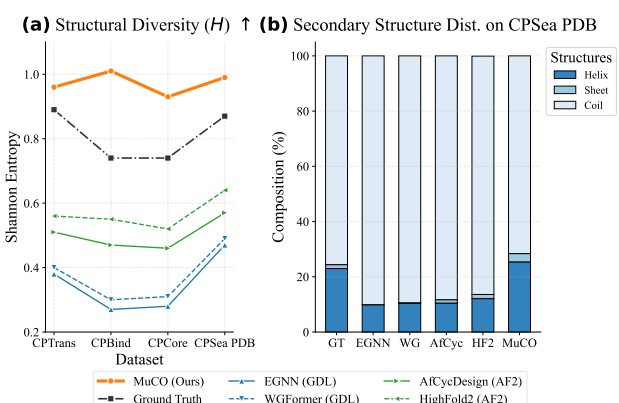

*Figure 4.* (a) Secondary Structural Diversity ($H_{SS}$) across different test sets; MuCO maintains superior ensemble diversity compared to all baselines. (b) Secondary structure composition (Helix, Sheet, and Coil) on the CPSea-PDB dataset. MuCO significantly improves the recovery of regular secondary structures.

**Evaluation of Physicochemical Properties.** To further assess the practical utility of MuCO in a drug discovery context, we expand our evaluation to key physicochemical properties that are critical determinants of metabolic stability and membrane permeability. As shown in Table 2, we evaluate the Hydrogen bond counts (H-bond), radius of gyration ($R_g$, Å), and solvent-accessible surface area (SASA, Å$^2$) using mean deviation and RMSD against the ground truth on the CPSea-PDB dataset. MuCO consis-

tently achieves the lowest deviations and RMSD across all properties, demonstrating that it generates highly compact, stable, and biologically relevant conformations compared to baselines.

*Table 2.* Evaluation of physicochemical properties on CPSea-PDB. Lower deviation and RMSD indicate generated structures are physically and pharmacologically closer to the native states.

| Method | H-bond | | $R_g$ (Å) | | SASA (Å$^2$) | |
|---|---|---|---|---|---|---|
| | Dev.↓ | RMSD↓ | Dev.↓ | RMSD↓ | Dev.↓ | RMSD↓ |
| EGNN | 2.82 | 4.32 | 0.54 | 0.74 | 144.0 | 189.7 |
| WGFormer | 3.17 | 4.07 | 0.67 | 0.72 | 178.3 | 190.9 |
| AfCycDesign | 3.98 | 4.94 | 0.56 | 0.87 | 266.6 | 302.3 |
| HighFold2 | 4.65 | 4.84 | 0.59 | 0.78 | 272.2 | 279.9 |
| **MuCO (Ours)** | **1.43** | **3.60** | **0.28** | **0.65** | **93.6** | **123.0** |

**Evaluation of Distributional Fidelity.** Furthermore, we evaluate the distributional fidelity of the generated ensembles by calculating the Jensen-Shannon (JS) similarity between the generated and ground-truth distributions for backbone torsions, side-chain torsions ($\chi_1$ and $\chi_2$), and secondary structures (Table 3). MuCO significantly outperforms both AF2-based and GDL-based baselines in all JS metrics. This highlights the advantage of our generative multi-stage framework in accurately capturing the underlying conformational distributions of cyclic peptides, rather than collapsing into deterministic, rigid states.

*Table 3.* Distributional fidelity on CPSea-PDB. JS Sim. stands for Jensen-Shannon Similarity against the ground truth (higher is better).

| Method | Backbone JS↑ | Side-chain JS↑ | Sec-Struct JS↑ |
|---|---|---|---|
| EGNN | 0.373 | 0.276 | 0.756 |
| WGFormer | 0.383 | 0.281 | 0.752 |
| AfCycDesign | 0.512 | 0.422 | 0.780 |
| HighFold2 | 0.571 | 0.462 | 0.804 |
| **MuCO** | **0.589** | **0.479** | **0.852** |

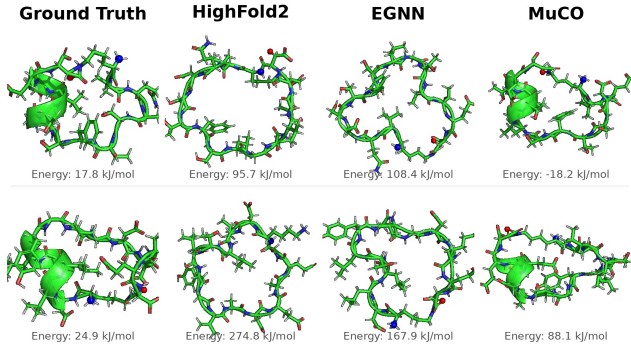

*Figure 5.* **Case study on structural fidelity.** Visual comparison of predicted conformations for a representative sample from the CPSea PDB dataset. Under single sampling, MuCO reconstructs the native-like $\alpha$-helical motif and achieves significantly lower potential energy ($E$) compared to representative baselines.

### 4.3. Analysis of Sampling Strategy

**Efficiency of Hierarchical Exploration.** Beyond the single sampling mode, MuCO can accelerate the exploration of the cyclic conformation space by the proposed hierarchical parallel sampling strategy. In Figure 6, we analyze the impact of increasing the sampling budget ($K$ Backbone states $\times$ $M$ Side-chain states). We observe a significant improvement in effectiveness: increasing the sampling budget drastically improves the cyclization mode coverage and achieves **a success rate of 100%**. Higher sampling density allows MuCO to discover more feasible cyclization modes given the same sequence (e.g. $> 90\%$ coverage for both H2T and K2DE), a capability completely absent in deterministic prediction models like HighFold2.

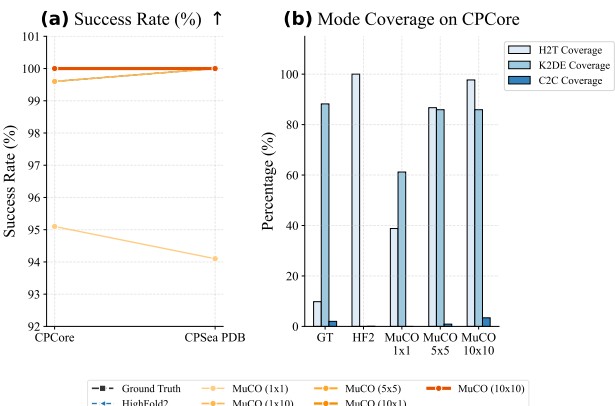

*Figure 6.* (a) Success rate as a function of the number of backbone ($K$) and side-chain ($M$) samples. (b) Cyclization mode coverage (H2T and K2DE) on the CPCore dataset. Increasing sampling density allows MuCO to discover feasible cyclization modes.

In particular, in a sample budget of $10 \times 10$, MuCO achieves a mean energy of -3.9 kJ/mol in CPCore and -247.6 kJ/mol in CPSea PDB, which surpasses the stability of Ground Truth conformations as shown in Figure 7. In addition, the amortized sampling time per conformation is reduced to the millisecond scale ($\approx 41$ms) as shown in Table 4, detailed sampling strategies and results are provided in Appendix E.

*Table 4.* Comparison of model complexity and inference efficiency. **MuCO (Parallel)** achieves the lowest amortized latency per sample ($K \times M = 100$) while maintaining a significantly smaller parameter footprint than folding-based methods.

| Method | Trainable Params ↓ | Amortized Time ↓ |
|---|---|---|
| AfCycDesign | 92.3M | 4236ms |
| HighFold2 | 92.3M | 4512ms |
| WGFormer | 78.8M | 103ms |
| EGNN | **1.5M** | 72ms |
| **MuCO (Parallel)** | 40.2M | **41ms** |

*Note:* Amortized time for MuCO is measured over 100 parallel samples. Our method is $\sim 1.7\times$ faster than the lightweight EGNN and $> 100\times$ faster than folding-based baselines.

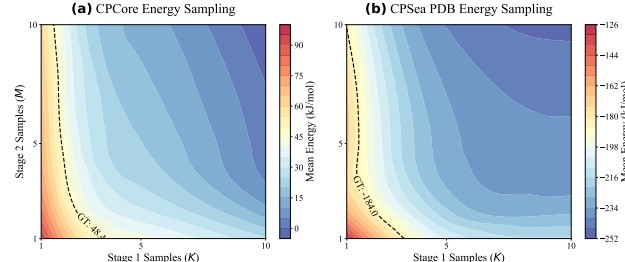

*Figure 7.* The heatmaps display the mean potential energy for (a) CPCore and (b) CPSea PDB with Stage-1 (backbone generation) and Stage-2 (side-chain packing) sample sizes. MuCO is capable of identifying metastable states with energies comparable to or lower than the ground truth with efficient parallel sampling.

### 4.4. Ablation Studies

**Necessity of Generative Modeling.** We validate our multi-stage generative modeling strategy in Figure 8. In particular, replacing the flow-based backbone generator of Stage-1 with an EGNN leads to a significant drop in success rate and diversity, confirming that the flow matching is essential to generate valid cyclic topologies. Similarly, replacing the flow-based side-chain packing module of Stage-2 with an EGNN results in higher energies, indicating that deterministic packing cannot resolve the dense steric constraints of macro-cycles as well as the generative method does. These results demonstrate the rationality of our model design.

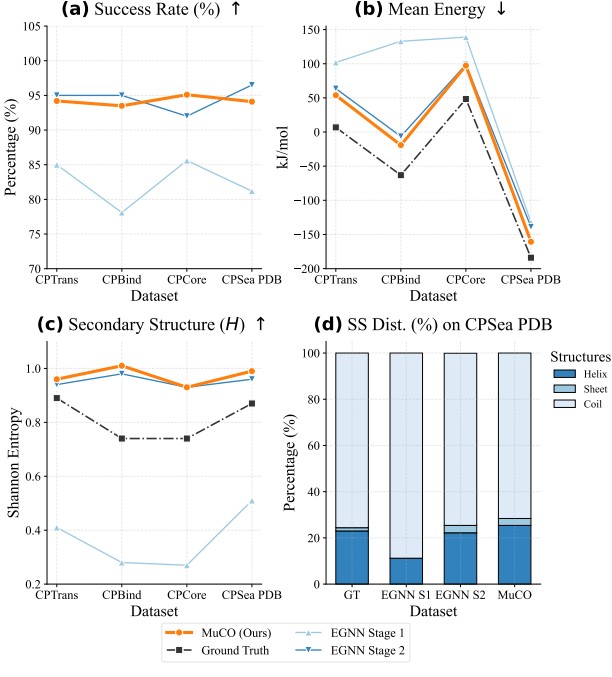

*Figure 8.* Comparison between the full MuCO model and the variants with EGNN-based deterministic Stage-1 or Stage-2. (a-b) Success rate and mean energy; (c-d) Secondary structure diversity and distribution.

**Necessity of Cyclic Relative Positional Encoding.** To isolate the contribution of our proposed Cyclic RPE in Stage-2 (Remark 2), we compare the side-chain generation quality of MuCO trained with standard linear RPE versus Cyclic RPE. Inspired by recent cyclic design works, we evaluate the Jensen-Shannon (JS) similarity against the ground truth for mean value of 4 individual side-chain torsions ($\chi$ JS) and their joint distribution ($\chi_1$-$\chi_2$ JS). As shown in Table 5, the Cyclic RPE captures local geometric information about cyclic peptides more effectively. Furthermore, different types of RPE distinguish linear and cyclic chains naturally.

*Table 5.* Ablation of Cyclic RPE on side-chain torsion JS similarities on CPSea-PDB.

| Ablation Setting | $\chi$ JS $\uparrow$ | $\chi_1$-$\chi_2$ JS $\uparrow$ |
|---|---|---|
| MuCO (Linear RPE) | 0.8867 | 0.4779 |
| **MuCO (Cyclic RPE)** | **0.8932** | **0.4794** |

**Necessity of Physics-Aware Optimization.** Figure 9 provides a deep dive into the role of Stage-3. An intriguing finding is that the raw MuCO output (w/o Opt) is dominated by random coils (Helix $\approx 0\%$), representing a poorly regulated but topologically valid potential well. Stage-3 acts as a potent folding driver, condensing these coils into well-defined helices and sheets. In contrast, HighFold2 shows minimal change after optimization, suggesting that its predictions are rigid and trapped in high-energy local minima.

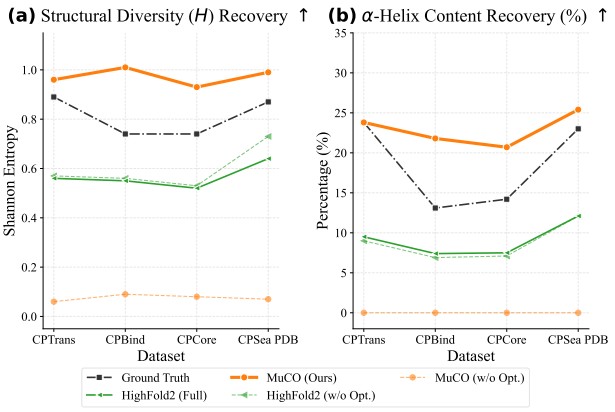

*Figure 9.* (a) Recovery of structural diversity ($H_{SS}$) before and after optimization. (b) Comparison of $\alpha$-helix content recovery against Ground Truth and baselines. Stage-3 refines stochastically sampled coordinates around potential wells into physically realizable local minima and well-defined secondary structural motifs.

**Generative Initialization vs. Random Seeding.** To determine whether the biologically relevant structures emerge entirely from the CHARMM36 optimization (Stage-3) or are genuinely guided by our generative models (Stages 1 and 2), we conduct a control experiment using random coordinate initialization followed by CHARMM36 minimization

on CPSea-PDB (20 independent trials). As shown in Table 6, random initialization fails to generate valid secondary structures during relaxation ($\Delta$SS $< 0$). In contrast, MuCO provides meaningful conformational guidance, achieving a high secondary structure gain ($\Delta$SS $= 0.2759$). This confirms that the force field can only recover native-like motifs when seeded with high-quality initial conformations generated by Stages 1 and 2.

*Table 6.* Secondary Structure (SS) ratio before and after CHARMM36 optimization. $\Delta$SS indicates the gain in secondary structure.

| Condition | SS Before | SS After | $\Delta$SS (Gain) |
|---|---|---|---|
| Random (First Sample) | 0.0314 | 0.0272 | -0.0042 |
| Random (Average of 20) | 0.0436 | 0.0194 | -0.0241 |
| Random (Best of 20) | 0.0436 | 0.0796 | 0.0360 |
| **MuCO** | **0.0080** | **0.2839** | **0.2759** |

The results of Figure 9 and Table 6 reveal that *MuCO's generative stages effectively seed the correct potential well, allowing physics-based optimization to efficiently descend into low-energy states corresponding to valid conformations.*

## 5. Conclusion and Future Work

In this work, we introduced MuCO, a multi-stage conformation optimization framework for generative peptide cyclization. Our method overcomes the mode collapse of existing methods by combining generative modeling with physics-aware optimization. Using an efficient hierarchical sampling strategy, MuCO explores the rugged energy landscape of cyclic peptides more effectively than deterministic prediction, achieving superior physical stability and structural diversity. By bridging the gap between static conformation prediction and dynamic physical reality, MuCO provides a computational tool for exploring the conformational space of macrocycles, potentially accelerating the design of peptide therapeutics.

**Limitations and Future Work.** Technically, our method focuses on monomeric peptides with terminal-based cyclizations, and its backbone folding efficiency degrades in ultra-long sequences. In addition, from a practical drug discovery perspective, current evaluations still lack comprehensive metrics on pharmaceutical properties and synthesizability. Therefore, future work will aim to design more physically meaningful model architectures and extend MuCO to multi-chain complexes and more diverse data sources. We are planning to collaborate with biologists on wet-lab verification and data annotation, paving the way for broader and more practical applications in drug design.

## Acknowledgements

This work was proposed by Hongteng Xu at Renmin University of China and was partially done during the internships of Yitian and Fanmeng at DP Technology. It was funded in part by the Beijing Municipal Science and Technology Commission, the Administrative Commission of Zhongguancun Science Park (No. Z251100007525009), the Beijing Major Science and Technology Project (No. Z251100008425002), and Fundamental and Interdisciplinary Disciplines Breakthrough Plan of the Ministry of Education of China (No. JYB2025XDXM702). We also acknowledge the support provided by the fund for building world-class universities (disciplines) at Renmin University of China, as well as by funds from the Beijing Key Laboratory of Research on Large Models and Intelligent Governance and the Engineering Research Center of Next-Generation Intelligent Search and Recommendation, Ministry of Education.

## Impact Statement

This paper presents MuCO, a generative framework designed to advance the field of macro-cyclic drug discovery and computational biology. Cyclic peptides are a critical class of therapeutics capable of targeting undruggable protein-protein interactions; however, their development has long been hindered by the difficulty of accurate conformational modeling. By providing an efficient tool for sampling diverse, low-energy ensembles, our work has the potential to accelerate the design of more stable and permeable peptide drugs, ultimately benefiting healthcare and therapeutic development. While we recognize that advanced molecular design tools carry a theoretical risk of being repurposed for harmful substances, we believe the positive impact on public health and scientific research far outweighs these concerns. We encourage the responsible use of these methodologies within the established frameworks of chemical and biological safety.

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

# A. Training Details

## A.1. Training Protocols and Hyperparameters

### A.1.1. STAGE-1: BACKBONE FLOW MATCHING

The backbone generation module is built on the FoldFlow-2 architecture (Huguet et al., 2024), employing a tripartite structure of an encoder, a multi-modal fusion trunk, and a geometric decoder. Each component is stacked with 2 blocks of Invariant Point Attention (IPA) or transformer layers. To integrate sequence information, we utilize a frozen ESM2-650M model, projecting its output into 128-dimensional sequential and 128-dimensional pair representations. In the subsequent combiner module, these are fused into a joint latent space with sequential and pair dimensions of 128 and 64. The model operates on the $SE(3)$ manifold with a coordinate scaling factor of $0.1$ to normalize the translational variance.

The training objective $\mathcal{L}_{\text{backbone}}$ is a multi-task composite loss that penalizes errors in flow prediction and structural geometry. We define total loss as a weighted sum of five components:

$$\mathcal{L}_{\text{backbone}} = \mathcal{L}_{\text{trans}} + 0.5\mathcal{L}_{\text{rot}} + \mathcal{L}_{\text{bb\_atom}} + \mathcal{L}_{\text{dist\_mat}} + 0.25\mathcal{L}_{\text{aux}}, \tag{5}$$

where $\mathcal{L}_{\text{trans}}$ and $\mathcal{L}_{\text{rot}}$ are flow matching losses for the translational and rotational components. To ensure local and global structural consistency, we introduce backbone atom loss $\mathcal{L}_{\text{bb\_atom}}$ and distance matrix loss $\mathcal{L}_{\text{dist\_mat}}$, both of which are activated for diffusion times $t > 0.25$. Finally, an auxiliary loss $\mathcal{L}_{\text{aux}}$ is used to penalize deviations in bond lengths and angles, ensuring that the generated scaffolds are physically reliable before entering the packing stage.

### A.1.2. STAGE-2: SIDE-CHAIN FLOW MATCHING

The side-chain packing module leverages the EquiformerV2 architecture to model cyclic macrocycles (Liao et al., 2024; Lee & Kim, 2025). The network consists of 4 equivariant layers with 256 sphere channels and 256 edge channels. We set the maximum degree of spherical harmonics at $L_{\max} = 3$ and the azimuthal quantum number at $M_{\max} = 1$. The attention mechanism is configured with 8 heads and a hidden dimension of 64. The input node features (95-dim) include one-hot amino acid identities and backbone torsion angles $(\psi, \tau, \omega)$, while the edge features (261-dim) combine $E(3)$-invariant geometric features with our proposed cyclic relative positional encodings to account for the ring-closed topology.

The model is trained using a Torsional Flow Matching objective on the hypertorus $\mathbb{T}^{4L}$. The training loss is defined as:

$$\mathcal{L}_{\text{packing}} = \mathbb{E}_{t,\boldsymbol{\chi}_1,\boldsymbol{\chi}_t} \left[ w(t) \| v_\tau^{\mathcal{C}}(\boldsymbol{\chi}_t, t, \mathcal{B}, \mathcal{S}) - u_t^{\mathcal{C}}(\boldsymbol{\chi}_t \mid \boldsymbol{\chi}_1) \|^2 \right], \tag{6}$$

where $w(t) = (1 - t)^{-2}$ is a weighting factor that prioritizes precision as the trajectory approaches the clean data state $(t \to 1)$. We adopt the velocity field formulation to optimize the flow. For residues with inherent $\pi$-symmetry, such as Phenylalanine or Tyrosine, the loss is computed using a mod $\pi$ operation on the torsion angles to prevent the model from being penalized for predicting symmetry-equivalent rotameric states.

## A.2. Hardware and Other Training Details

All models were trained on four NVIDIA RTX 4090 GPUs (24GB VRAM each) and an Intel(R) Xeon(R) Gold 6348 CPU (112 logic cores, 2.60GHz). For Stage-1 (Backbone), each GPU consumed approximately 16GB of VRAM. Due to the requirement of high-throughput data loading for large-scale backbone ensembles, the CPU utilization remained near maximum capacity during this stage. Stage-2 (Side-chain) was more memory-efficient, utilizing roughly 12GB of VRAM per GPU. Both stages were implemented in PyTorch and optimized using the AdamW optimizer.

We utilized the CPSea dataset as our data source and partitioned it into training set and validation set. During the training process, we observed that the backbone flow matching module converges with relative stability, but the side-chain packing module fluctuates severely, likely due to the highly constrained and rugged energy landscape of dense macrocycles.

For training configurations. For Stage-1, the model was trained for 100 epochs with a learning rate of $2 \times 10^{-4}$ and a total batch size of 256. Each epoch took approximately 12 minutes to complete, totaling roughly 20 hours of training time. For Stage-2, we set the learning rate at $1 \times 10^{-4}$ and the batch size at 128. This stage was also trained for 100 epochs, each epoch taking approximately 8 minutes, resulting in a total training duration of approximately 13 hours. We utilized an exponential moving average (EMA) with a decay rate of 0.999 to stabilize the training dynamics. To ensure the robustness of the flow matching, a velocity field loss was adopted with a tolerance $\epsilon = 2 \times 10^{-4}$ during training.

In particular, in our baselines, EGNN and WGFormer were trained on CPSea using the corresponding linear peptide conformations as structural priors. Conversely, for AF2-based models, we utilized their pre-trained weights without further fine-tuning. This is justified by the fact that the CPSea dataset (excluding the PDB subset) is largely derived from AF2 predictions; thus, these models can be considered to have prior exposure to the underlying data distribution.

### A.3. Algorithm

We present the training procedure for our decoupled flow matching modules in Algorithm 1.

---

**Algorithm 1** Training of MuCO Decoupled Flow Modules

---

1: **Input:** Dataset $\mathcal{D} = \{(\mathcal{S}, \boldsymbol{X})\}$ (Sequences $\mathcal{S}$ and Conformations $\boldsymbol{X}$)
2: **Parameters:** Backbone Flow Network $v_\theta^B$, Side-chain Flow Network $v_\tau^{\mathcal{C}}$
3: **Hyperparameters:** Time distribution $p(t) = \mathcal{U}[0, 1]$
4: **while** not converged **do**
5:     Sample batch $(\mathcal{S}, \boldsymbol{X}) \sim \mathcal{D}$
6:     Decompose $\boldsymbol{X}$ into Backbone frames $\mathcal{B}_1$ and Side-chain torsions $\mathcal{C}_1$
7:     *// — Stage-1: Backbone Flow Matching Training —*
8:     Sample time $t \sim p(t)$
9:     Sample prior $\mathcal{B}_0 \sim p_{\text{prior}}^{\mathcal{B}}$ {Gaussian on $\mathbb{R}^3$, Isotropic on SO(3)}
10:     Compute geodesic path $\mathcal{B}_t$ between $\mathcal{B}_0$ and $\mathcal{B}_1$ on $(\text{SE}(3) \times \mathbb{T})^L$
11:     Compute target vector field $u_t^{\mathcal{B}}(\mathcal{B}_t \mid \mathcal{B}_1)$
12:     $\mathcal{L}_{\text{backbone}} \leftarrow \mathcal{L}_{\text{trans}} + 0.5\mathcal{L}_{\text{rot}} + \mathcal{L}_{\text{bb\_atom}} + \mathcal{L}_{\text{dist\_mat}} + 0.25\mathcal{L}_{\text{aux}}$
13:     Update $\theta$ by minimizing $\mathcal{L}_{\text{backbone}}$
14:     *// — Stage-2: Side-chain Flow Training —*
15:     Sample time $t \sim p(t)$
16:     Sample prior $\mathcal{C}_0 \sim \mathcal{U}(\mathbb{T}^{4L})$ {Uniform on Torus}
17:     Compute geodesic path $\mathcal{C}_t$ between $\mathcal{C}_0$ and $\mathcal{C}_1$ on $\mathbb{T}^{4L}$
18:     Compute target vector field $u_t^C(\mathcal{C}_t \mid \mathcal{C}_1)$
19:     $\mathcal{L}_{\text{packing}} \leftarrow \mathbb{E}_{t, \boldsymbol{\chi}_1, \boldsymbol{\chi}_t} \left[ w(t) \| v_\tau^{\mathcal{C}}(\boldsymbol{\chi}_t, t, \mathcal{B}, \mathcal{S}) - u_t^{\mathcal{C}}(\boldsymbol{\chi}_t \mid \boldsymbol{\chi}_1) \|^2 \right]$
20:     Update $\tau$ by minimizing $\mathcal{L}_{\text{packing}}$
21: **end while**

---

## B. Rule-based Topology Detection and Physics Refinement

To bridge the gap between the probabilistic samples generated by flow matching stages and the particular physical constraints required for downstream applications, we implemented an automated rule-based topology detection and refinement strategy. This strategy ensures that the generated conformations strictly adhere to specific chemical bonding criteria and facilitates energy minimization to find potential wells automatically.

### B.1. Implicit Constraint Subspaces

We define the associated implicit constraint subspaces $\Omega_{\mathcal{C}}$ for the three most prevalent cyclization strategies in the CPSea dataset. Let $\boldsymbol{x}_{i,\text{atom}}$ denote the Cartesian coordinates of a specific atom in residue $i$. We define the target equilibrium peptide bond length as $d_{\text{pep}} = 1.33$ Å and the disulfide bond length as $d_{\text{SS}} = 2.05$ Å. A tolerance $\delta = 0.1$ Å is applied to verify successful closure.

**Head-to-Tail Cyclization ($\Omega_{\textbf{H2T}}$):** Closure is achieved by forming a peptide bond between the N-terminus of the first residue ($i = 1$) and the C-terminus of the last residue ($i = L$):

$$\Omega_{\text{H2T}} = \{\boldsymbol{X} \mid \|\boldsymbol{x}_{1,\text{N}} - \boldsymbol{x}_{L,\text{C}}\|_2 \in [d_{\text{pep}} - \delta, d_{\text{pep}} + \delta]\}. \tag{7}$$

**Disulfide Bridging** ($\Omega_{\text{SS}}^{(i,j)}$)**:** A covalent bond forms between the sulfur atoms ($\text{S}^\gamma$) of two cysteine residues at positions $i$ and $j$:

$$\Omega_{\text{SS}}^{(i,j)} = \left\{ X \;\middle|\; \begin{array}{l} \|\boldsymbol{x}_{i,\text{S}^\gamma} - \boldsymbol{x}_{j,\text{S}^\gamma}\|_2 \in [d_{\text{SS}} - \delta, d_{\text{SS}} + \delta], \\ \text{where } s_i = \text{Cys}, s_j = \text{Cys}. \end{array} \right\} \tag{8}$$

**Side-chain Isopeptide Linkage** ($\Omega_{\text{Iso}}^{(i,j)}$)**:** An amide bond forms between a Lys side-chain amine ($\text{N}^\zeta$) and an Asp/Glu side-chain carboxylate ($\text{C}^\gamma$ for Asp or $\text{C}^\delta$ for Glu):

$$\Omega_{\text{Iso}}^{(i,j)} = \left\{ X \;\middle|\; \begin{array}{l} \|\boldsymbol{x}_{i,\text{N}^\zeta} - \boldsymbol{x}_{j,\text{C}^{\delta/\gamma}}\|_2 \in [d_{\text{pep}} - \delta, d_{\text{pep}} + \delta], \\ \text{where } s_i = \text{Lys}, s_j \in \{\text{Asp, Glu}\}. \end{array} \right\} \tag{9}$$

### B.2. Topology Inference and Force Field Optimization

**Prioritized Inference Logic:** Since the cyclization mode is not explicitly labeled during inference, we adopt a priority-based detection strategy. The algorithm first identifies all Cys-Cys pairs; if the distance between any pair's $\text{S}^\gamma$-$\text{S}^\gamma$ is less than 2.5 Å, it is assigned as a disulfide bridge. If no such pair exists but potential isopeptide residues (Lys, Asp/Glu) are present, we compare the current distance of the closest isopeptide pair with the head-to-tail distance, selecting the shorter of the two. In all other cases, the peptide is treated as head-to-tail cyclic.

**Force Field Patching:** We utilize the CHARMM36 force field (Huang & MacKerell Jr, 2013) for local geometry refinement. Standard residue templates are insufficient for cyclic bonds because of atom loss. We implemented custom patches to handle these transitions: the Cys template removes sulfur hydrogen (HG); the Lys template removes two hydrogens from the $\text{N}^\zeta$ amine; the Asp/Glu templates remove the hydroxyl group (OH) from the side-chain carboxylate; and the backbone H2T mode performs standard dehydration.

**Validation and Success Criteria:** After 500 steps of Steepest Descent (SD) minimization, we perform a final geometric check. A conformation is only considered valid if its minimized bond length $d_{\text{final}}$ satisfies $|d_{\text{final}} - d_{\text{target}}| \leq \delta$. Samples that fail this physics-based threshold are discarded. Given that our datasets focus on single-mode cyclization, this automated heuristic effectively resolves most of the cases.

### B.3. Algorithm

---
**Algorithm 2** Rule-based Topology Detection and Physics Refinement

---
1: **Input:** Raw coordinates $\widetilde{X}$, Sequence $S$, Targets $d_{\text{pep}}, d_{\text{SS}}$, Tolerance $\delta$
2: **Output:** Refined coordinates $X$ or *Failure*
3: Identify reactive pairs $\mathcal{P} \leftarrow \{(i,j) \mid \text{Cys-Cys, Lys-Asp/Glu, 1-}L\}$
4: **if** $\exists(i,j) \in \text{Cys-Cys}$ **and** $\|\boldsymbol{x}_{i,\text{S}^\gamma} - \boldsymbol{x}_{j,\text{S}^\gamma}\|_2 < 2.5\text{Å}$ **then**
5: $\quad$ Mode $\leftarrow \Omega_{\text{SS}}^{(i,j)}$
6: **else if** $\exists(i,j) \in$ Isopeptide candidates **then**
7: $\quad$ Mode $\leftarrow (\|\text{Iso-pair}\|_2 < \|\text{H2T-pair}\|_2)?\Omega_{\text{Iso}}^{(i,j)} : \Omega_{\text{H2T}}$
8: **else**
9: $\quad$ Mode $\leftarrow \Omega_{\text{H2T}}$
10: **end if**
11: Apply CHARMM36 patches (remove leaving atoms) and minimize $\widetilde{X} \rightarrow X$
12: **if** $|d_{\text{final}} - d_{\text{target}}| \leq \delta$ **then**
13: $\quad$ **return** $X$
14: **else**
15: $\quad$ **return** *Failure*
16: **end if**

---

# C. Dataset and Data Processing

To ensure MuCO learns from high-quality and structurally representative cyclic peptides, we curated a large-scale dataset derived from the CPSea repository (Yang et al., 2026). This section details our filtering criteria and the statistical characteristics of the resulting subsets.

## C.1. Filtering Pipeline

Our filtering process is designed to isolate peptides where the macro-cyclic constraint is the primary determinant of the conformational landscape. First, we select peptides with lengths $L \in [8, 16]$, a range that is most critical for therapeutic applications. Second, we introduce a *Backbone Mass Ratio* filter (referred to as Residue Summary). We calculate the relative molecular mass of the cyclic backbone and ensure that it exceeds 20% of the total molecular weight (ResSum) in the multi-chain conformation. This ensures that the training conformations are not biased by external chain environments, preserving intrinsic low-energy states that authentically reflect the geometric influence of the ring-closure.

## C.2. Dataset Subsets and Topology

As illustrated in Figure 10, we partition the filtered CPSea dataset into five disjoint subsets based on their functional annotations and data sources: **CPSea**: The full set of filtered cyclic peptides (∼75k samples, Training set). **CPBind**: Peptides with high binding affinity. **CPTrans**: Cell-penetrating peptides. **CPCore**: The intersection of CPBind and CPTrans, representing the most biologically relevant core. **CPSea PDB**: A gold-standard subset consisting of high-resolution structures directly sourced from the Protein Data Bank (PDB).

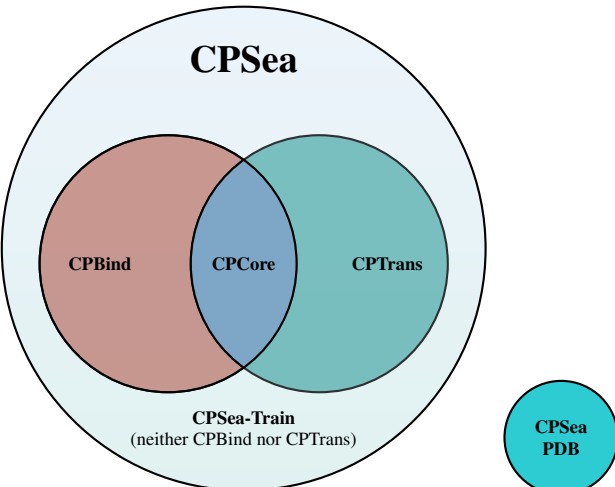

*Figure 10.* Venn diagram illustrating the relationships between different CPSea subsets. CPSea PDB serves as an external high-confidence validation set.

## C.3. Data Statistics and Distributions

We provide a comprehensive statistical summary of the sequence and structural features across all subsets in Table 7. The distributions are further visualized in Figure 11.

**Sequence Characteristics.** The sequence length distribution across all subsets is negatively skewed (Skewness ≈ −1.3), with a heavy concentration at $L = 16$ (see Figure 11, left). The mean length ranges from 14.2 to 15.2. The *Residue Summary* (Backbone Mass Ratio) KDE plots (Figure 11, middle) show a sharp, unimodal peak at approximately 0.22 with extremely low variance (Std < 0.03), indicating high structural homogeneity across the filtered dataset.

**Cyclization Modes.** As shown in the Ring Info distribution (Figure 11, right), the dataset covers three main cyclization strategies: Stapled (Isopeptide, K-to-DE), Disulfide (Cys-to-Cys), and Head-to-tail. The stapled peptides are the most numerous in the total CPSea set, and the frequency of Stapled configurations is particularly high in the CPCore and CPSea PDB subsets (exceeding 80%). This highlights the importance of side-chain cross-linking in high-confidence cyclic structures. Despite the vast difference in sample sizes (ranging from 85 in CPSea PDB to more than 75,000 in CPSea) and their functions, the frequency trends remain remarkably consistent across all subsets.

*Table 7.* Descriptive statistics of sequence length ($L$) and backbone mass ratio (ResSum) for CPSea subsets.

| Category | Dataset | Count | Mean $L$ | Skew ($L$) | Mean ResSum | Skew (ResSum) |
|---|---|---|---|---|---|---|
| Training | CPSea (Train) | 75,337 | 14.99 | -1.28 | 0.2327 | 1.56 |
| Testing | CPTrans | 6,590 | 14.23 | -0.66 | 0.2300 | 1.68 |
| | CPBind | 3,194 | 15.05 | -1.52 | 0.2289 | 1.68 |
| | CPCore | 263 | 14.40 | -0.80 | 0.2273 | 1.59 |
| | CPSea PDB | 85 | 15.25 | -1.75 | 0.2228 | 0.98 |

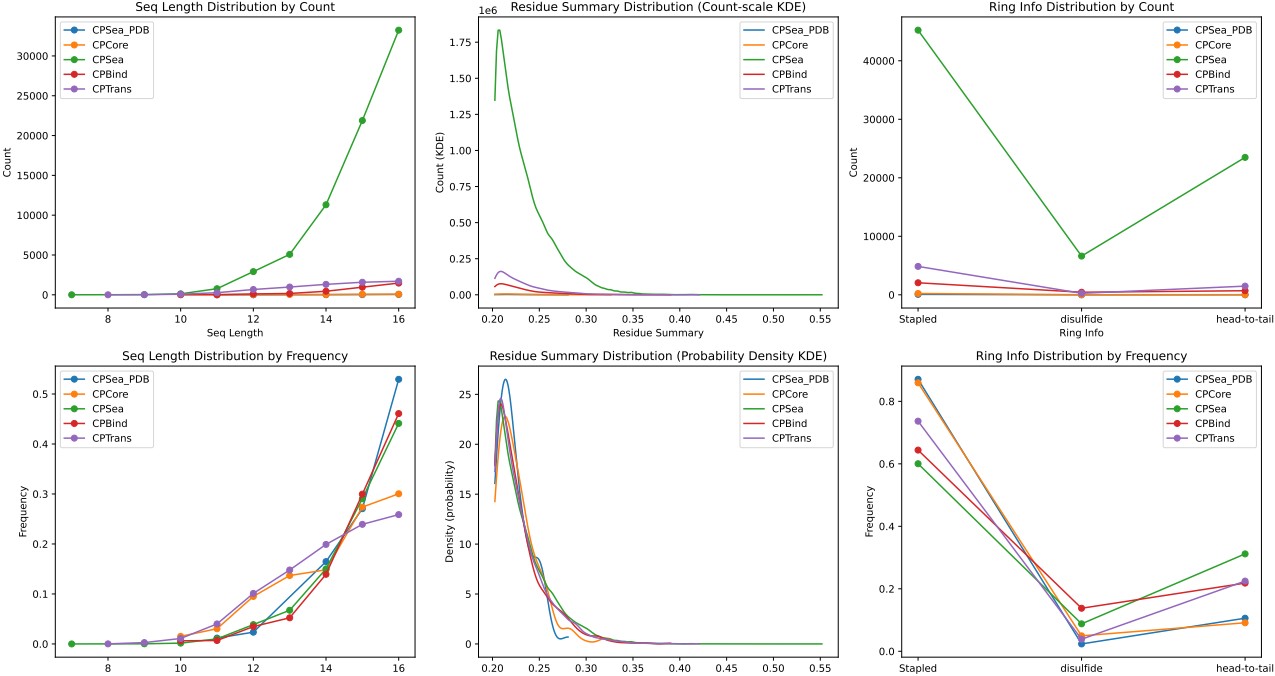

*Figure 11.* Distributions of sequence length, backbone mass ratio (Residue Summary), and cyclization modes across the five subsets. Top row shows absolute counts; bottom row shows normalized frequencies.

# D. Detailed Evaluation Metrics

This section describes the metrics used to assess the generative quality of MuCO. To ensure an equitable comparison, we implement a **unified evaluation protocol**: all candidate structures, including MuCO generations, baseline predictions, and the original Ground Truth (GT) samples, are processed through the identical force-field optimization pipeline described in Appendix B before any metrics are calculated.

## D.1. Structural Diversity and Distributional Fidelity

We employ Shannon Entropy ($H = -\sum_{i=1}^{n} p_i \ln p_i$) to evaluate distributional diversity in **single-sampling mode**, specifically reporting $H_{\mathcal{C}}$ for cyclization modes and $H_{\text{SS}}$ for secondary structures. Furthermore, to verify if the model accurately captures the true conformational landscape, we quantify **distributional fidelity** using Jensen-Shannon (JS) similarity, defined as JS $\text{Sim}(P, Q) = 1 - \frac{1}{2} [D_{\text{KL}}(P \parallel M) + D_{\text{KL}}(Q \parallel M)]$, where $M = \frac{1}{2}(P + Q)$ and $D_{\text{KL}}$ is the Kullback-Leibler divergence. $P$ and $Q$ denote the generated and ground-truth probability distributions, respectively. We compute this metric for 1D distributions (e.g., backbone and side-chain torsions), and extend it to 2D joint distributions $P(\chi_1, \chi_2)$ for side-chain torsions and $P(\alpha\text{-helix\%}, \beta\text{-helix\%})$ for secondary structures to evaluate the preservation of coupled local geometries.

### D.2. Physical Stability and Success Rate

The Success Rate (SR) is the percentage of structures satisfying the geometric closure tolerance $\delta$ (defined in Appendix B) with a final potential energy below $10^3$ kJ/mol. Conformations exceeding this energy threshold are categorized as cyclization failures, as they typically exhibit severe steric clashes and physical instability. Mean Potential Energy ($E$, kJ/mol) is calculated via the CHARMM36 force field. By minimizing all structures (including baselines and GT) prior to measurement, we fairly evaluate each model's ability to provide a high-quality initial seed that reliably relaxes into a stable local minimum.

### D.3. Secondary Structure and Physicochemical Properties

To evaluate local folding fidelity, we use the `mdtraj` library to assign secondary structures via the DSSP algorithm, classifying residues into Helix ($\alpha$), Sheet ($\beta$), and Coil. To further assess the practical utility in drug discovery, we evaluate key **physicochemical properties**: Hydrogen bond counts, radius of gyration ($R_g$), and solvent-accessible surface area (SASA). These metrics, critical for evaluating membrane permeability and metabolic stability, are measured using mean deviation and RMSD against the ground truth to verify structural compactness and biological relevance.

## E. Efficiency and Complexity Analysis

This section provides a detailed analysis of the computational complexity and inference performance of MuCO. We compare its multi-stage generative architecture against end-to-end folding models and geometric graph baselines to highlight its advantages in large-scale ensemble generation.

### E.1. Model Parameter Comparison

As shown in Table 8, MuCO has a competitive trainable parameter count of 40.2M (22.1M for Stage-1 and 18.0M for Stage-2). Although Stage-1 incorporates a frozen ESM2-650M model, the sequence embeddings are pre-computed and cached, effectively removing the language model's overhead from the iterative sampling loop. AF2-based variants (AfCycDesign and HighFold2) involve 92.3M trainable parameters; despite the absence of MSAs for shorter peptides, their high-dimensional transformer blocks impose significant memory and latency costs. In particular, while EGNN has the fewest parameters (1.5M), it exhibits disproportionately high training and inference latency, requiring approximately 90 minutes per epoch. This is likely due to the computational overhead of iterative coordinate-dependent message passing, which lacks the vectorized efficiency of the attention based mechanisms utilized in MuCO and WGFormer.

### E.2. Sampling Speed and Parallel Efficiency

The inference efficiency of MuCO is a critical advantage, particularly when modeling the dynamic conformational ensembles required for macro-cyclic drug design. As demonstrated in Table 8, we evaluate the latency across two dimensions: the initialization (loading) phase and the active inference phase. While AF2-based models are typically deterministic, we treat the outputs from their five standard parameter sets as 5 independent samples to evaluate their search efficiency, while GDL-based model only has 1 single sample (sample size).

**The Loading Bottleneck** A significant drawback of AfCycDesign and HighFold2 is their extensive model-loading overhead, which exceeds 50 seconds on a single NVIDIA RTX 4090 GPU. This delay is primarily due to the heavy weights of the Evoformer blocks and the complex initialization of the ColabFold-optimized platform. In contrast, MuCO's total loading time is approximately 14.6 seconds. Although Stage-1 incorporates a large ESM2-650M language model, its impact on latency is mitigated by our caching strategy: sequence embeddings are computed once and stored in a high-speed buffer, allowing the SE(3) flow matcher to function with a throughput comparable to lightweight EGNN.

**Serial vs. Parallel Throughput** In a conventional serial inference setting (sampling one conformation at a time), MuCO's total latency is 1219 ms—roughly four times faster than AF2-based models but slower than regressive GDLs like WGFormer. However, the true strength of MuCO's decoupled architecture lies in its *tree-structured parallel sampling* capability.

Taking advantage of batch-processing, Stage-1 can generate a batch of $N = 10$ diverse backbone scaffolds in 1312 ms. Subsequently, Stage-2 populates these scaffolds by sampling $M$ side-chain configurations in parallel. For instance, populating 100 packing configurations across these 10 backbones requires only 2742 ms in Stage-2. Consequently, the total

wall-clock time to generate a high-quality ensemble of 100 physically refined conformations is:

$$T_{\text{total}} = T_{\text{backbone}}(\text{batch} = 10) + T_{\text{packing}}(\text{batch} = 100) \approx 4054 \text{ ms}. \tag{10}$$

This translates to an **amortized latency of 40.5 ms per sample**. In practical terms, while an AF2-based model is still loading its weights or predicting its first static structure ($\sim$4.5 s), MuCO can deliver a fully optimized ensemble of 100 diverse low-energy states. Although specific times could vary with server I/O and CPU load, the trend underscores MuCO's superior scalability for potential well screening for macro-cycles.

*Table 8.* Comparison of model complexity and inference latency on a single RTX 4090 GPU. Parallel latency for MuCO reflects the total time to generate 100 samples ($N = 10, M = 10$). Parallel MuCO sampling reaches the lowest amortized time generating a candidate conformation.

| Method | Trainable Params | Loading Time | Sample Size | Sampling Time | Amortized Time |
|---|---|---|---|---|---|
| AfCycDesign | 92.3M | 53.9 s | 5 | 21182 ms | 4236 ms |
| HighFold2 | 92.3M | 52.7 s | 5 | 22561 ms | 4512 ms |
| WGFormer | 78.8M | 7.3 s | 1 | 103 ms | 103 ms |
| EGNN | **1.5M** | **6.4 s** | 1 | **72 ms** | 72 ms |
| MuCO (Serial) | 40.2M | 14.6 s | 1 | 1219 ms | 1219 ms |
| **MuCO (Parallel)** | 40.2M | 14.6 s | **100** | 4054 ms | **41 ms** |

## E.3. Algorithm

We present our hierarchical inference strategy in Algorithm 3.

---

**Algorithm 3** Inference: Hierarchical $K \times M$ Sampling & Optimization

---

1: **Input:** Sequence $\mathcal{S}$, Backbone Samples $K$, Packing Samples $M$
2: **Output:** Optimized Conformation Ensemble $\boldsymbol{X}_{\text{final}}$
3: {// — Stage-1: Topology Generation ($K$ parallel samples) —}
4: Initialize $\mathcal{B}_0^{(i)} \sim p_{\text{prior}}^B$ for $i = 1 \ldots K$
5: **for** $k = 0$ **to** $K_{\text{steps}} - 1$ **do**
6: $\quad t_k \leftarrow k/K_{\text{steps}}$
7: $\quad \mathcal{B}^{(i)} \leftarrow v_\theta^B(\mathcal{B}_{t_k}^{(i)}, t_k, \mathcal{S})$ {Batched inference}
8: $\quad \mathcal{B}_{t_{k+1}}^{(i)} \leftarrow \text{Step}(\mathcal{B}_{t_k}^{(i)}, \mathcal{B}^{(i)}, \Delta t)$
9: **end for**
10: Set Backbones $\{\mathcal{B}_1^{(i)}\}_{i=1}^K$
11: {// — Stage-2: Conditional Packing ($K \times M$ parallel samples) —}
12: Initialize $\mathcal{C}_0^{(i,j)} \sim \mathcal{U}(\mathbb{T}^{4L})$ for $i = 1 \ldots K, j = 1 \ldots M$
13: **for** $k = 0$ **to** $K_{\text{steps}} - 1$ **do**
14: $\quad t_k \leftarrow k/K_{\text{steps}}$
15: $\quad$ {Condition flow on corresponding backbone $i$ with Cyclic RPE}
16: $\quad \mathcal{C}^{(i,j)} \leftarrow v_\tau^{\mathcal{C}}(\mathcal{C}_{t_k}^{(i,j)}, t_k | \mathcal{B}_1^{(i)}, \mathcal{S})$
17: $\quad \mathcal{C}_{t_{k+1}}^{(i,j)} \leftarrow \mathcal{C}_{t_k}^{(i,j)} + \mathcal{C}^{(i,j)} \cdot \Delta t$
18: **end for**
19: Reconstruct Atoms $\widetilde{\boldsymbol{X}}^{(i,j)} \leftarrow \text{ForwardKinematics}(\mathcal{B}_1^{(i)}, \mathcal{C}_1^{(i,j)})$
20: {// — Stage-3: Physics-Aware Optimization —}
21: **for** each conformation $\boldsymbol{X}^{(i,j)}$ **do**
22: $\quad \mathcal{T} \leftarrow \text{DetectTopology}(\widetilde{\boldsymbol{X}}^{(i,j)})$ {Algorithm2}
23: $\quad \boldsymbol{X}^{(i,j)} \leftarrow \text{Minimize}(\widetilde{\boldsymbol{X}}^{(i,j)}, \mathcal{E}_{\text{CHARMM36}}, \mathcal{T})$
24: **end for**
25: **return** Best $\boldsymbol{X}^{(i,j)}$ based on energy $E$

---

# F. More Experimental Details and Results

## F.1. Detailed Experimental Results

*Table 9.* Quantitative comparison across four datasets. MuCO consistently outperforms baselines, achieving the lowest potential energy ($E$) and maintaining high structural diversity ($H$) comparable to the ground truth.

*Dataset: **CPTrans***

| Method | Succ. | Energy | Div. | Modes (%) | | |
|---|---|---|---|---|---|---|
| | (%) | (kJ/mol) | ($H$) | H2T | K2DE | C2C |
| GT (Ref.) | - | 6.8 | 0.57 | 23.5 | 74.2 | 2.3 |
| EGNN | 87.6 | 126.3 | 0.57 | 24.3 | 74.2 | 1.5 |
| WGFormer | 88.2 | 122.5 | 0.58 | 22.1 | 76.5 | 1.4 |
| AfCycDesign | 98.5 | 142.5 | 0.01 | 99.9 | 0.1 | 0.0 |
| HighFold2 | **98.9** | 138.3 | 0.00 | 99.9 | 0.0 | 0.0 |
| **MuCO** | 94.2 | **53.9** | **0.69** | 45.6 | 54.1 | 0.3 |

*Dataset: **CPBind***

| Method | Succ. | Energy | Div. | Modes (%) | | |
|---|---|---|---|---|---|---|
| | (%) | (kJ/mol) | ($H$) | H2T | K2DE | C2C |
| GT (Ref.) | - | -63.2 | 0.63 | 23.9 | 67.7 | 8.3 |
| EGNN | 80.6 | 70.8 | 0.59 | 20.0 | 76.0 | 4.0 |
| WGFormer | 81.5 | 68.4 | 0.61 | 21.5 | 73.8 | 4.7 |
| AfCycDesign | 98.0 | 61.2 | 0.00 | 99.8 | 0.2 | 0.0 |
| HighFold2 | **98.7** | 54.4 | 0.00 | 99.9 | 0.0 | 0.0 |
| **MuCO** | 93.5 | **-19.2** | **0.68** | 57.2 | 41.7 | 1.1 |

*Dataset: **CPCore***

| Method | Succ. | Energy | Div. | Modes (%) | | |
|---|---|---|---|---|---|---|
| | (%) | (kJ/mol) | ($H$) | H2T | K2DE | C2C |
| GT (Ref.) | - | 48.4 | 0.36 | 9.8 | 88.2 | 2.0 |
| EGNN | 91.6 | 164.1 | 0.40 | 11.3 | 87.5 | 1.2 |
| WGFormer | 92.1 | 160.2 | 0.42 | 12.5 | 86.0 | 1.5 |
| AfCycDesign | 99.0 | 180.5 | 0.00 | 100.0 | 0.0 | 0.0 |
| HighFold2 | **99.6** | 174.3 | 0.00 | 100.0 | 0.0 | 0.0 |
| **MuCO** | 95.1 | **97.3** | **0.67** | 40.8 | 58.8 | 0.4 |

*Dataset: **CPSea PDB***

| Method | Succ. | Energy | Div. | Modes (%) | | |
|---|---|---|---|---|---|---|
| | (%) | (kJ/mol) | ($H$) | H2T | K2DE | C2C |
| GT (Ref.) | - | -184.0 | 0.54 | 11.9 | 86.9 | 1.2 |
| EGNN | 92.9 | -63.1 | 0.35 | 12.7 | 87.3 | 0.0 |
| WGFormer | 91.8 | -58.5 | 0.41 | 14.1 | 85.9 | 0.0 |
| AfCycDesign | 98.8 | -35.5 | 0.00 | 100.0 | 0.0 | 0.0 |
| HighFold2 | **100.0** | -43.2 | 0.00 | 100.0 | 0.0 | 0.0 |
| **MuCO** | 94.1 | **-160.7** | **0.88** | 30.0 | 70.0 | 0.0 |

*Note:* **Succ.**: Success Rate; **H2T**: Head to Tail; **C2C**: Disulfide; **K2DE**: Isopeptide. Lower energy indicates better stability.

*Table 10.* Secondary structure composition analysis. MuCO significantly improves the recovery of $\alpha$-helices and $\beta$-sheets compared to GNN and AlphaFold2-based baselines. By arranging datasets side-by-side, we observe consistent performance across all tasks.

*Dataset: **CPTrans***

| Method | $\alpha$ | $\beta$ | Coil | $H$ |
|---|---|---|---|---|
| GT (Ref.) | 23.8 | 1.6 | 74.7 | 0.89 |
| EGNN | 7.4 | 0.0 | 92.6 | 0.38 |
| WGFormer | 7.8 | 0.1 | 92.1 | 0.40 |
| AfCycDesign | 8.5 | 1.3 | 90.2 | 0.51 |
| **MuCO** | **23.8** | **2.8** | **73.4** | **0.96** |

*Dataset: **CPBind***

| Method | $\alpha$ | $\beta$ | Coil | $H$ |
|---|---|---|---|---|
| GT (Ref.) | 13.1 | 2.8 | 84.0 | 0.74 |
| EGNN | 4.6 | 0.0 | 95.4 | 0.27 |
| WGFormer | 5.1 | 0.2 | 94.7 | 0.30 |
| AfCycDesign | 6.2 | 2.1 | 91.7 | 0.47 |
| **MuCO** | **21.8** | **4.7** | **73.5** | **1.01** |

*Dataset: **CPCore***

| Method | $\alpha$ | $\beta$ | Coil | $H$ |
|---|---|---|---|---|
| GT (Ref.) | 14.2 | 2.3 | 83.4 | 0.74 |
| EGNN | 4.9 | 0.0 | 95.1 | 0.28 |
| WGFormer | 5.5 | 0.1 | 94.4 | 0.31 |
| AfCycDesign | 6.8 | 1.5 | 91.7 | 0.46 |
| **MuCO** | **20.7** | **3.4** | **75.9** | **0.93** |

*Dataset: **CPSea PDB***

| Method | $\alpha$ | $\beta$ | Coil | $H$ |
|---|---|---|---|---|
| GT (Ref.) | 23.0 | 1.4 | 75.6 | 0.87 |
| EGNN | 9.9 | 0.0 | 90.1 | 0.47 |
| WGFormer | 10.5 | 0.1 | 89.4 | 0.49 |
| AfCycDesign | 10.5 | 1.2 | 88.3 | 0.57 |
| **MuCO** | **25.4** | **3.0** | **71.6** | **0.99** |

*Note:* Values represent mean percentage (%). Headers abbreviated: $\alpha$=Helix, $\beta$=Sheet. Diversity ($H$) uses Shannon Entropy.

*Table 11.* Impact of sampling strategy on MuCO performance. We evaluate MuCO across different sampling budgets ($K \times M$), where $K$ and $M$ denote the number of backbone scaffolds and side-chain packing configurations, respectively. Increasing the budget significantly lowers the potential energy and improves success rates, enabling MuCO to discover multiple feasible cyclization modes simultaneously.

| Method (Config.) | Success | Mean Energy ($E_{\text{final}}$) | | Mode Coverage (%) | | |
|---|---|---|---|---|---|---|
| | (%) | (kJ/mol) | (eV) | H2T | K2DE | C2C |
| *Dataset: CPCore* | | | | | | |
| GT (Ref.) | - | 48.4 | 0.50 | 9.8 | 88.2 | 2.0 |
| HighFold2 | 99.6 | 174.3 | 1.81 | **100.0** | 0.0 | 0.0 |
| MuCO ($1 \times 1$) | 95.1 | 96.3 | 1.00 | 38.8 | 61.2 | 0.0 |
| MuCO ($1 \times 10$) | 99.6 | 61.9 | 0.64 | 40.8 | 63.4 | 0.0 |
| MuCO ($5 \times 5$) | **100.0** | 20.1 | 0.21 | 86.7 | 85.9 | 1.9 |
| MuCO ($10 \times 1$) | **100.0** | 26.0 | 0.27 | 94.3 | 85.9 | 1.9 |
| **MuCO ($10 \times 10$)** | **100.0** | **-3.9** | **-0.04** | 97.7 | **85.9** | **3.4** |
| *Dataset: CPSea PDB* | | | | | | |
| GT (Ref.) | - | -184.0 | -1.91 | 11.9 | 86.9 | 1.2 |
| HighFold2 | **100.0** | -43.2 | -0.45 | **100.0** | 0.0 | 0.0 |
| MuCO ($1 \times 1$) | 94.1 | -126.8 | -1.32 | 33.8 | 66.3 | 0.0 |
| MuCO ($1 \times 10$) | **100.0** | -184.6 | -1.92 | 37.6 | 70.6 | 0.0 |
| MuCO ($5 \times 5$) | **100.0** | -231.4 | -2.41 | 85.9 | 87.1 | 0.0 |
| MuCO ($10 \times 1$) | **100.0** | -214.9 | -2.23 | 89.4 | 87.1 | 0.0 |
| **MuCO ($10 \times 10$)** | **100.0** | **-247.6** | **-2.57** | 94.1 | **87.1** | **0.0** |

*Note:* Mode Coverage indicates the percentage of targets for which at least one valid conformation of that mode was generated. Values may sum to $> 100\%$ for high sampling budgets, indicating the model successfully recovered multiple feasible modes (e.g., both H2T and K2DE) for the same peptide sequences.

*Table 12.* Ablation study of MuCO components. We compare the full framework against variants removing Stage 1 (generative backbone) or Stage 2 (flow-based packing). Stage 1 is crucial for success and diversity, while Stage 2 is critical for energy minimization.

*Dataset: CPTrans*

| Method | Succ. | Energy | Div. | Modes (%) | | |
|---|---|---|---|---|---|---|
| | (%) | (kJ/mol) | ($H$) | H2T | K2DE | C2C |
| GT (Ref.) | - | 6.8 | 0.57 | 23.5 | 74.2 | 2.3 |
| HighFold2 | **98.9** | 138.3 | 0.00 | 99.9 | 0.0 | 0.0 |
| EGNN Stage 1 | 85.0 | 101.9 | 0.66 | 36.7 | 62.8 | 0.5 |
| EGNN Stage 2 | 95.0 | 63.6 | 0.69 | 52.0 | 47.9 | 0.1 |
| **MuCO** | 94.2 | **53.9** | **0.69** | 45.6 | 54.1 | 0.3 |

*Dataset: CPBind*

| Method | Succ. | Energy | Div. | Modes (%) | | |
|---|---|---|---|---|---|---|
| | (%) | (kJ/mol) | ($H$) | H2T | K2DE | C2C |
| GT (Ref.) | - | -63.2 | 0.63 | 23.9 | 67.7 | 8.3 |
| HighFold2 | **98.7** | 54.4 | 0.00 | 99.9 | 0.0 | 0.0 |
| EGNN Stage 1 | 78.1 | 132.9 | 0.65 | 44.0 | 48.0 | 8.0 |
| EGNN Stage 2 | 95.0 | -6.4 | 0.65 | 65.3 | 34.1 | 0.6 |
| **MuCO** | 93.5 | **-19.2** | **0.68** | 57.2 | 41.7 | 1.1 |

*Dataset: CPCore*

| Method | Succ. | Energy | Div. | Modes (%) | | |
|---|---|---|---|---|---|---|
| | (%) | (kJ/mol) | ($H$) | H2T | K2DE | C2C |
| GT (Ref.) | - | 48.4 | 0.36 | 9.8 | 88.2 | 2.0 |
| HighFold2 | **99.6** | 174.3 | 0.00 | 100.0 | 0.0 | 0.0 |
| EGNN Stage 1 | 85.6 | 139.0 | 0.61 | 29.8 | 70.2 | 0.0 |
| EGNN Stage 2 | 92.0 | 98.8 | **0.69** | 44.2 | 55.4 | 0.4 |
| **MuCO** | 95.1 | **97.3** | 0.67 | 40.8 | 58.8 | 0.4 |

*Dataset: CPSea PDB*

| Method | Succ. | Energy | Div. | Modes (%) | | |
|---|---|---|---|---|---|---|
| | (%) | (kJ/mol) | ($H$) | H2T | K2DE | C2C |
| GT (Ref.) | - | -184.0 | 0.54 | 11.9 | 86.9 | 1.2 |
| HighFold2 | **100.0** | -43.2 | 0.00 | 100.0 | 0.0 | 0.0 |
| EGNN Stage 1 | 81.2 | -132.0 | 0.67 | 30.4 | 68.1 | 1.4 |
| EGNN Stage 2 | 96.5 | -138.9 | **0.97** | 40.2 | 59.8 | 0.0 |
| **MuCO** | 94.1 | **-160.7** | 0.88 | 30.0 | 70.0 | 0.0 |

*Note:* **Succ.**: Success Rate; **Div.**: Diversity. "EGNN Stage 1/2" use regressive EGNN in corresponding stages. The full MuCO model balances stability and diversity best.

*Table 13.* Ablation analysis of secondary structure components. Removing Stage 1 (Backbone) causes collapse into random coils, while removing Stage 2 (FlowPacking) yields suboptimal distributions. MuCO achieves the most balanced profile, matching the ground truth diversity ($H$).

<table>
<tr><td colspan="5" align="center">Dataset: CPTrans</td></tr>
<tr><td>Method</td><td>$\alpha$</td><td>$\beta$</td><td>Coil</td><td>$H$</td></tr>
<tr><td>GT (Ref.)</td><td>23.8</td><td>1.6</td><td>74.7</td><td>0.89</td></tr>
<tr><td>EGNN Stage 1</td><td>8.2</td><td>0.0</td><td>91.8</td><td>0.41</td></tr>
<tr><td>EGNN Stage 2</td><td>21.6</td><td>3.0</td><td>75.4</td><td>0.94</td></tr>
<tr><td>MuCO</td><td>23.8</td><td>2.8</td><td>73.4</td><td>0.96</td></tr>
</table>

<table>
<tr><td colspan="5" align="center">Dataset: CPBind</td></tr>
<tr><td>Method</td><td>$\alpha$</td><td>$\beta$</td><td>Coil</td><td>$H$</td></tr>
<tr><td>GT (Ref.)</td><td>13.1</td><td>2.8</td><td>84.0</td><td>0.74</td></tr>
<tr><td>EGNN Stage 1</td><td>4.9</td><td>0.0</td><td>95.1</td><td>0.28</td></tr>
<tr><td>EGNN Stage 2</td><td>19.7</td><td>4.8</td><td>75.4</td><td>0.98</td></tr>
<tr><td>MuCO</td><td>21.8</td><td>4.7</td><td>73.5</td><td>1.01</td></tr>
</table>

<table>
<tr><td colspan="5" align="center">Dataset: CPCore</td></tr>
<tr><td>Method</td><td>$\alpha$</td><td>$\beta$</td><td>Coil</td><td>$H$</td></tr>
<tr><td>GT (Ref.)</td><td>14.2</td><td>2.3</td><td>83.4</td><td>0.74</td></tr>
<tr><td>EGNN Stage 1</td><td>4.5</td><td>0.1</td><td>95.5</td><td>0.27</td></tr>
<tr><td>EGNN Stage 2</td><td>19.2</td><td>3.8</td><td>77.0</td><td>0.93</td></tr>
<tr><td>MuCO</td><td>20.7</td><td>3.4</td><td>75.9</td><td>0.93</td></tr>
</table>

<table>
<tr><td colspan="5" align="center">Dataset: CPSea PDB</td></tr>
<tr><td>Method</td><td>$\alpha$</td><td>$\beta$</td><td>Coil</td><td>$H$</td></tr>
<tr><td>GT (Ref.)</td><td>23.0</td><td>1.4</td><td>75.6</td><td>0.87</td></tr>
<tr><td>EGNN Stage 1</td><td>11.2</td><td>0.0</td><td>88.8</td><td>0.51</td></tr>
<tr><td>EGNN Stage 2</td><td>22.2</td><td>3.2</td><td>74.5</td><td>0.96</td></tr>
<tr><td>MuCO</td><td>25.4</td><td>3.0</td><td>71.6</td><td>0.99</td></tr>
</table>

*Note:* Columns represent percentage (%) of $\alpha$-Helix, $\beta$-Sheet, and Coil. $H$: Diversity (Shannon Entropy). "EGNN Stage 1/2" use regressive EGNN in corresponding stages. The full MuCO model balances stability and diversity best.

*Table 14.* Impact of force field optimization on secondary structure. We compare structural composition before (w/o Opt.) and after refinement. MuCO exhibits a dramatic transition from disordered raw states to regular structures during optimization, whereas HighFold2 (HF2) remains largely static.

<table>
<tr><td colspan="5" align="center">Dataset: CPTrans</td></tr>
<tr><td>Method</td><td>$\alpha$</td><td>$\beta$</td><td>Coil</td><td>$H$</td></tr>
<tr><td>GT (Ref.)</td><td>23.8</td><td>1.6</td><td>74.7</td><td>0.89</td></tr>
<tr><td>HF2 (w/o Opt.)</td><td>9.0</td><td>1.9</td><td>89.1</td><td>0.57</td></tr>
<tr><td>HF2 (Final)</td><td>9.5</td><td>1.5</td><td>89.0</td><td>0.56</td></tr>
<tr><td>MuCO (w/o Opt.)</td><td>0.0</td><td>0.7</td><td>99.3</td><td>0.06</td></tr>
<tr><td>MuCO (Final)</td><td>23.8</td><td>2.8</td><td>73.4</td><td>0.96</td></tr>
</table>

<table>
<tr><td colspan="5" align="center">Dataset: CPBind</td></tr>
<tr><td>Method</td><td>$\alpha$</td><td>$\beta$</td><td>Coil</td><td>$H$</td></tr>
<tr><td>GT (Ref.)</td><td>13.1</td><td>2.8</td><td>84.0</td><td>0.74</td></tr>
<tr><td>HF2 (w/o Opt.)</td><td>6.9</td><td>3.0</td><td>90.1</td><td>0.56</td></tr>
<tr><td>HF2 (Final)</td><td>7.4</td><td>2.5</td><td>90.1</td><td>0.55</td></tr>
<tr><td>MuCO (w/o Opt.)</td><td>0.0</td><td>1.2</td><td>98.8</td><td>0.09</td></tr>
<tr><td>MuCO (Final)</td><td>21.8</td><td>4.7</td><td>73.5</td><td>1.01</td></tr>
</table>

<table>
<tr><td colspan="5" align="center">Dataset: CPCore</td></tr>
<tr><td>Method</td><td>$\alpha$</td><td>$\beta$</td><td>Coil</td><td>$H$</td></tr>
<tr><td>GT (Ref.)</td><td>14.2</td><td>2.3</td><td>83.4</td><td>0.74</td></tr>
<tr><td>HF2 (w/o Opt.)</td><td>7.1</td><td>2.3</td><td>90.6</td><td>0.53</td></tr>
<tr><td>HF2 (Final)</td><td>7.5</td><td>1.9</td><td>90.7</td><td>0.52</td></tr>
<tr><td>MuCO (w/o Opt.)</td><td>0.0</td><td>1.0</td><td>99.0</td><td>0.08</td></tr>
<tr><td>MuCO (Final)</td><td>20.7</td><td>3.4</td><td>75.9</td><td>0.93</td></tr>
</table>

<table>
<tr><td colspan="5" align="center">Dataset: CPSea PDB</td></tr>
<tr><td>Method</td><td>$\alpha$</td><td>$\beta$</td><td>Coil</td><td>$H$</td></tr>
<tr><td>GT (Ref.)</td><td>23.0</td><td>1.4</td><td>75.6</td><td>0.87</td></tr>
<tr><td>HF2 (w/o Opt.)</td><td>12.1</td><td>3.1</td><td>84.8</td><td>0.73</td></tr>
<tr><td>HF2 (Final)</td><td>12.1</td><td>1.5</td><td>86.3</td><td>0.64</td></tr>
<tr><td>MuCO (w/o Opt.)</td><td>0.0</td><td>0.8</td><td>99.2</td><td>0.07</td></tr>
<tr><td>MuCO (Final)</td><td>25.4</td><td>3.0</td><td>71.6</td><td>0.99</td></tr>
</table>

*Note:* **HF2**: HighFold2. "w/o Opt." is the raw output before force field refinement; "Final" is after refinement. MuCO effectively folds disordered (high coil, low $H$) raw states into regular structures during optimization.

## F.2. Comparison with End-to-End Predictive Models

To further validate MuCO's generative framework, we benchmark it against two state-of-the-art end-to-end models, AlphaFold3 (Abramson et al., 2024) and ProteinZen (Li & Kortemme, 2025), on the CPSea-PDB dataset. As shown in Table 15, while AlphaFold3 achieves a high success rate (comparable to AfCycDesign and HighFold2), it still struggles to accurately recover secondary structures. In single-sampling mode, MuCO achieves significantly lower potential energy profiles and superior JS similarities for both torsions and secondary structures compared to these end-to-end predictive baselines.

*Table 15.* Comparison with AlphaFold3 and ProteinZen on CPSea-PDB.

| Method | Success (↑) | Energy (↓) | JS Similarity (↑) | | |
|---|---|---|---|---|---|
| | (%) | (kJ/mol) | Backbone | Side-chain | Sec-Struct |
| AlphaFold3 | **96.5** | -114.53 | 0.548 | 0.464 | 0.686 |
| ProteinZen | 49.4 | 454.76 | 0.416 | 0.326 | 0.700 |
| **MuCO** | 94.1 | **-160.71** | **0.589** | **0.479** | **0.852** |

*Note:* JS Similarity stands for Jensen-Shannon Similarity against the ground truth distributions (higher is better). Energy denotes the mean potential energy after physical refinement.

## F.3. Examples of Sampling

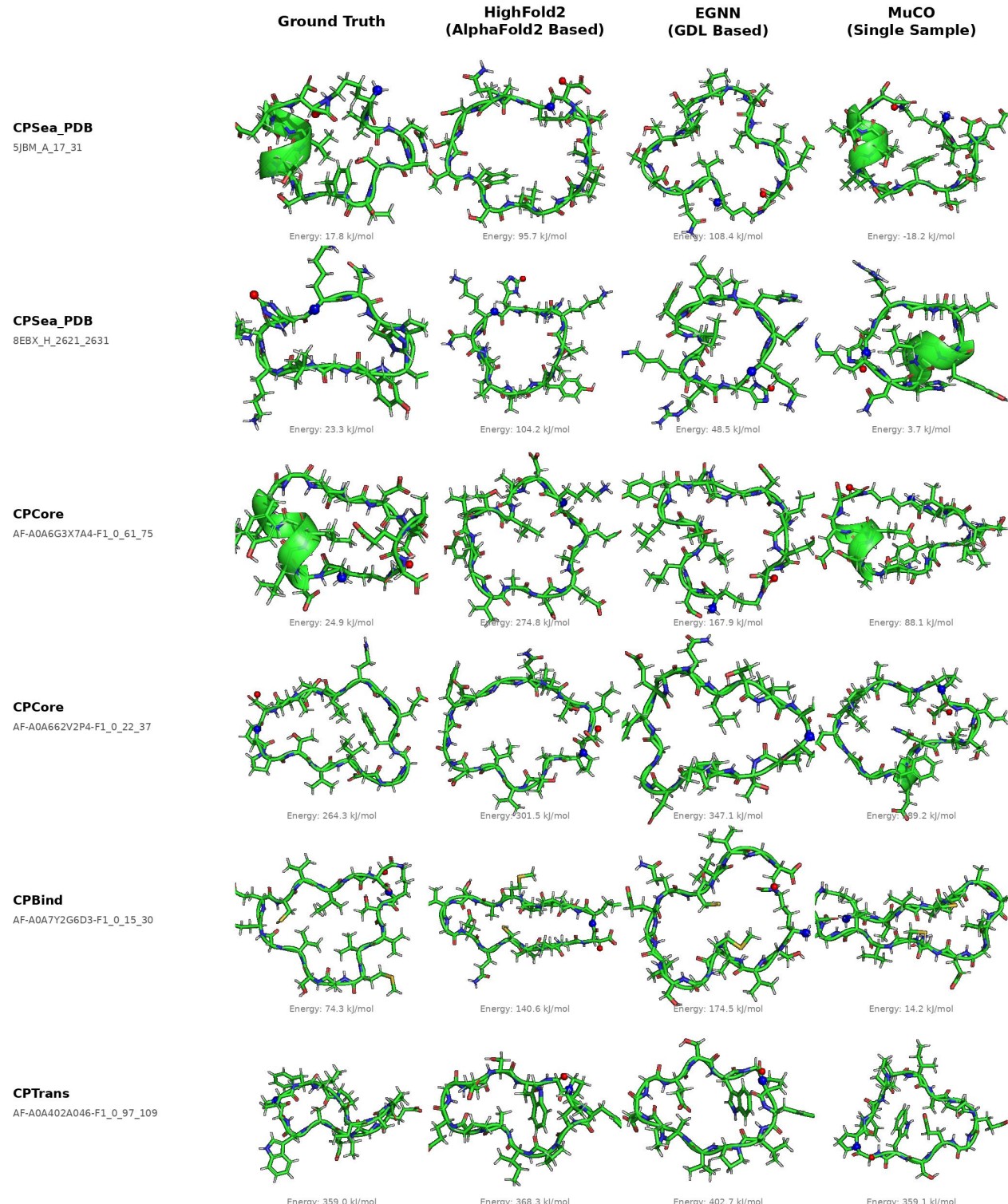

*Figure 12.* **Visual Comparison** of predicted cyclic peptide conformations across diverse sub-datasets. Even in **single sample mode**, MuCO consistently achieves significantly lower potential energy ($E$) compared to HighFold2 and EGNN baselines. Notably, MuCO demonstrates superior fidelity in recovering native-like **secondary structure motifs** (e.g., $\alpha$-helices), whereas the baseline models often collapse into high-energy, disordered states.

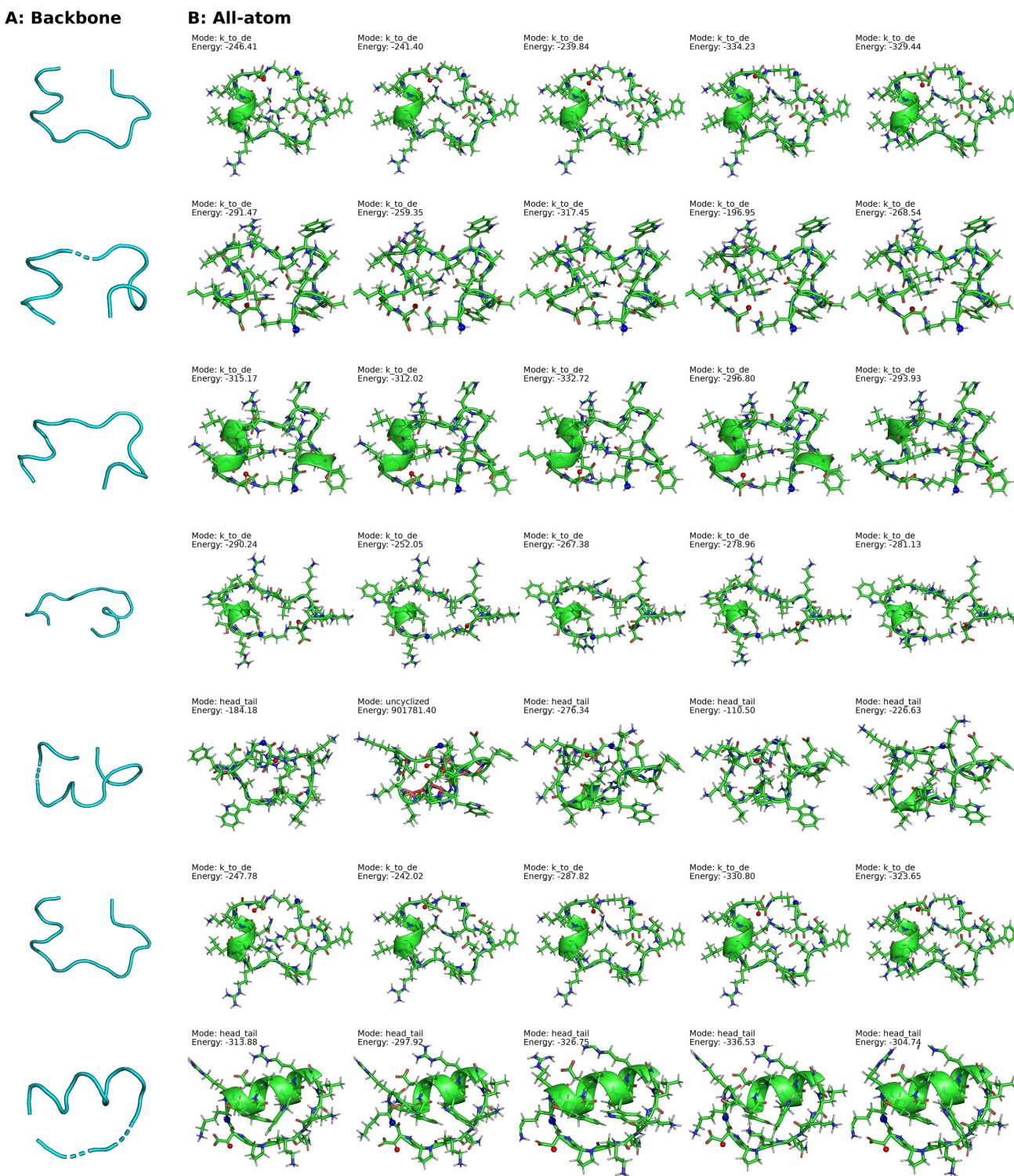

*Figure 13.* **Sample of *1JD6_A_16_31* generated by MuCO.** Column (A) shows the Stage-1 backbone scaffolds (cyan), while columns (B) show the corresponding all-atom samples (green) after Stage 2 and 3. The diversity of k-to-de and head-to-tail cycles highlights the model's capacity to sample multiple metastable states. Annotations specify the cyclization mode and potential energy (kJ/mol).

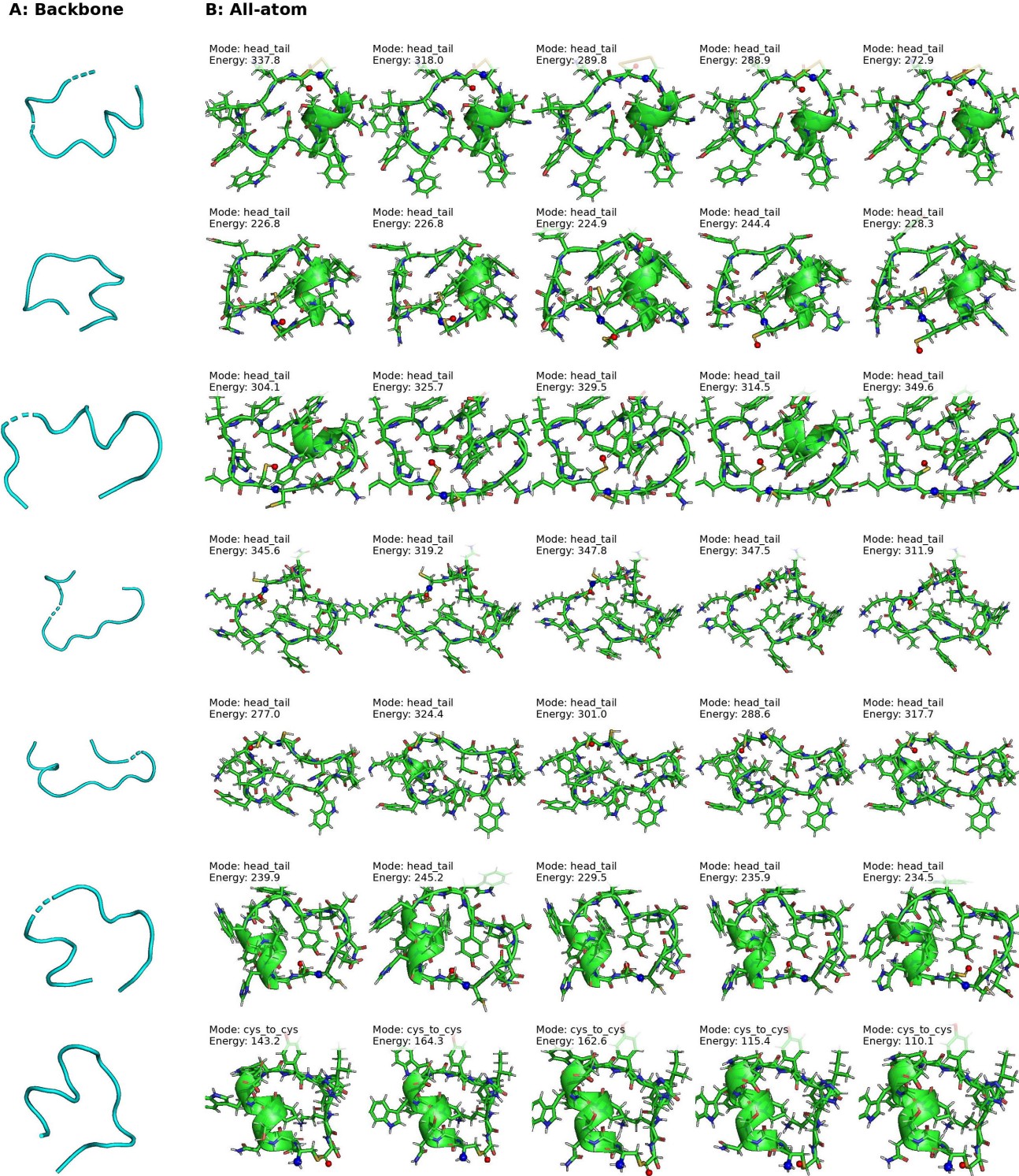

*Figure 14.* **Sample of *AF-X1M060-F1_0_15_29* generated by MuCO.** Column (A) shows the Stage-1 backbone scaffolds (cyan), while columns (B) show the corresponding all-atom samples (green) after Stage 2 and 3. The peptide structures demonstrate that MuCO effectively identifies and incorporates two distinct cyclization modes: head-to-tail backbone closure and cys-to-cys disulfide bridges.

