# OpenReview forum: "MuCO: Generative Peptide Cyclization Empowered by Multi-stage Conformation Optimization"
_ICML.cc/2026/Conference — ICML 2026 regular_

### Official Review · Reviewer_g9uf · 2026-03-07

**Soundness:** 3
**Presentation:** 4
**Significance:** 3
**Originality:** 3
**Overall Recommendation:** 5
**Confidence:** 4

**Summary:**

This paper tackles the problem of cyclic peptide conformation generation, given the linear sequence. The proposed method, MuCO, first samples different backbone conformations of the linear peptides, then identifies the cyclic-relevant conformation to add cyclic connections and generates side chains. Finally, the authors leverage openmm to refine the structures with physical force fields. The experiments are conducted on a large-scale dataset established by previous literature, and metrics of various aspects including cyclic success rates, physical energy, diversity, and distributions are calculated to comprehensively evaluate the models. Further analysis show that increasing number of initial conformation samples keep lowering down the energies, facilitating finding the most stable conformation.

**Compliance With Llm Reviewing Policy:**

Affirmed.

**Final Justification:**

The authors solved all my concerns, and also promise to fix the credit problem in the publication version. Thus I will keep my recommendation of acceptance.

**Key Questions For Authors:**

1. Is this algorithm extendable to protein-cyclic peptide complexes?
2. Is it generalizable to new cyclic methods beyond those mentioned in the paper?

**Limitations:**

yes

**Strengths And Weaknesses:**

**Strengths**

1. The paper is well written, with methods and experiments clearly presented. I can follow the points in the paper quickly without difficulty.
2. The motivation of leveraging generative sampling to bypass the rugged landscape is reasonable and interesting.
3. The proposed method is solid and sound, with comprehensive experiments and analysis to show the benefits of the proposed framework.

**Weaknesses**

1. Regarding the performance gain of increasing K*M in section 4.3, an ablation of the effect of conformation diversity on the energy could be added. Currently it is assumed that generative modeling could bypass the problems introduced by the rugged energy landscape and provide more stable conformations as initial conformations for later openmm optimizations. However, if the diversity of the generative model is low, it is possible that even with K*M sampling, the energy is not as good as reported in the paper. Therefore, it could be analyzed that whether the diversity of the generative model has much influence on the performance.

2. Currently the model is trained on cyclic peptide data, so it always output cyclic conformations. However, if actually there is no cyclic conformation for a linear sequence, the model will output hallucinations. How to address this problem?

---

> ### Author Rebuttal · Authors · 2026-03-29
>
> Thanks for your appreciation of our work. Please find our detailed responses to your specific concerns below.
>
> **Q1: Add an ablation study on the diversity vs. energy for the $K$-$M$  hierarchical sampling.**
>
> **A1:** The relationship between conformational diversity and energy performance has been analyzed in our paper. As shown in Figure 7, we display the heatmaps of the mean potential energy for CPCore and CPSea-PDB as a function of the sample sizes used in Stage 1 ($K$) and Stage 2 ($M$). As $K$ and $M$ increase, MuCO's ability to find low-energy conformations with diverse sampling improves.
>
> **Q2: Handling hallucinations for uncyclizable sequences.**
>
> **A2:** We agree that some sequences may be unstable as cyclic peptides. In MuCO, Stage 3 (Physics-Aware Optimization) serves as a physical validity filter. If a generated conformation is hallucinated and unstable, it will reach high potential energy or violate geometric constraints after Stage 3. **As shown in Appendix B.2 and D.2, the structures that exceed energy thresholds or bond tolerance criteria are classified as cyclization failures.**
>
> **Q3: Extendability to protein-cyclic peptide complexes.**
>
> **A3:** We acknowledge that the current MuCO framework focuses on the single-chain cyclic peptides. However, the backbone flow matching and side-chain packing modules can be adapted to multi-chain systems by extending the cyclic RPE to encompass inter-chain interactions. We are actively developing this extension to model multi-chain protein-cyclic peptide complexes, but this is out of scope for this work. We have noted this direction as our future work in Section 5.
>
> **Q4: Is it generalizable to new cyclic modes?**
>
> **A4:** Currently, MuCO can handle head-to-tail, disulfide, and isopeptide cyclizations. When we meet a new cyclic mode, our rule-based topology detection and CHARMM36 are capable of supporting a new linkage. In particular, Stage 3 of our method can refine conformations automatically as long as the definitions of geometric criteria and force-field parameters are provided.
>
> We appreciate your time and constructive comments. We hope that our responses can address all your concerns and make you more confident in supporting our work. We are open to any further questions you may have.

---

> > ### Author Rebuttal · Reviewer_g9uf · 2026-04-03
> >
> > Thanks for the responses. I will maintain my positive score. However, I would like to mention that after reading the provided codes, I found the implementation of the physical relaxer are from previous work (the CPSea dataset). So instead of claiming "our rule-based topology detection and CHARMM36", it is better to give the credit to whom it belongs. This should be made clear in the camera ready paper.

---

> > > ### Author Response · Authors · 2026-04-05
> > >
> > > Thank you for your continued support and positive evaluation. We appreciate you pointing out the credit attribution regarding the physical relaxer. The implementation of the physical relaxer is indeed built upon the open-source pipeline provided by the CPSea dataset and CP-Composer. We will rephrase statements and explicitly cite in the later version. Thank you again for your meticulous review.

---

### Official Review · Reviewer_A43d · 2026-03-07

**Soundness:** 3
**Presentation:** 3
**Significance:** 3
**Originality:** 3
**Overall Recommendation:** 4
**Confidence:** 4

**Summary:**

This paper proposes MuCO, a three-stage cyclic peptide generation framework.
Conditioned on a linear peptide, MuCO first designs the backbone confirmation, then packs the side chains, and finally performs physics-aware all-atom optimization.
Trained on the large-scale peptide dataset CPSea, MuCO achieves both high generation quality and high efficiency.

**Compliance With Llm Reviewing Policy:**

Affirmed.

**Key Questions For Authors:**

See "Weakness".

**Limitations:**

Yes

**Strengths And Weaknesses:**

**Strengths**
* The idea of three-stage generation reasonable and effective.
* The experimental results are good.
* The paper is well-written and easy to understand.

**Weaknesses**
* From a practical perspective, the evaluation metrics used are not very meaningful. Even if the generated cyclic peptides are physically stable and diverse, it does not mean they are useful. The authors may consider including experiments that can demonstrate MuCO's ability to design cyclic molecules with properties of interest, e.g., the ability to bind to a specific target.
* Based on current experiments, it remains unclear whether the good performance are mainly due to the MuCO framework itself or the CPSea dataset. Is it possible that a simpler cyclic peptide generation model (i.e., an end-to-end one) trained on CPSea can also achieve good performance?
* Real-world cyclic peptides often contain non-canonical amino acids, how can MuCO be extended to handle this?
* Does a three-stage framework suffer from severer error accumulation than an end-to-end one?
* What will MuCO do if it is fed with a linear peptide that cannot be cyclized stablely?

---

> ### Author Rebuttal · Authors · 2026-03-29
>
> Thanks for your appreciation of our work. Please find our detailed responses to your specific concerns below.
>
> **Q1: Consider pharmaceutical metrics and other metrics.**
>
> **A1:** To resolve your concern, we extended our evaluation on the distributions of backbone torsions, side-chain torsions ($\chi_1$ and $\chi_2$), and secondary structures. In particular, for each term, we consider the Jensen-Shannon (JS) similarity between the generated distribution and the ground truth. The results below demonstrate the advantage of MuCO.
>
> | Method | Success Rate | Energy (Mean) | Backbone JS Sim. | Side-chain JS Sim. | Sec-Struct JS Sim. |
> | :--- | :---: | :---: | :---: | :---: | :---: |
> | **EGNN** | 0.929 | -63.08 | 0.373 | 0.276 | 0.756 |
> | **WGFormer** | 0.918 | -58.53 | 0.383 | 0.281 | 0.752 |
> | **AfCycDesign** | 0.988 | -35.51 | 0.512 | 0.422 | 0.780 |
> | **HighFold2** | **1.000** | -43.24 | 0.571 | 0.462 | 0.804 |
> | **MuCO** | 0.941 | **-160.71** | **0.589** | **0.479** | **0.852** |
>
> We also extended evaluation on physicochemical properties, including H-bond (H-bond counts in cyclic peptide), $R_g$ (radius of gyration, $Å$), and SASA (solvent-accessible surface area, $Å^2$), and evaluated them on both mean deviation and RMSD. These metrics are essential for assessing the structural stability, compactness, and solubility of cyclic peptides, all of which are critical determinants of their metabolic stability and membrane permeability.
>
> |CPSea-PDB|H-bond Deviation|H-bond RMSD|$R_g$ Deviation|$R_g$ RMSD|SASA Deviation|SASA RMSD|
> | :--- | :---: | :---: | :---: | :---: | :---: | :---: |
> |**EGNN**|2.8228|4.3239|0.5395|0.7441|144.0284|189.6571|
> |**WGFormer**|3.1690|4.0704|0.6680|0.7192|178.2712|190.8699 |
> |**AfCycDesign**|3.9753|4.9383|0.5621|0.8723|266.6482|302.3426|
> |**HighFold2**|4.6471|4.8353|0.5936|0.7798|272.1592|279.9198|
> |**MuCO**|**1.4250**|**3.6030**|**0.2810**|**0.6479**|**93.6179**|**123.0153**|
>
> **Q2: Consider end-to-end models as baselines.**
>
> **A2:** To resolve your concern, we apply two state-of-the-art end-to-end models, AlphaFold3 and ProteinZen, as baselines, and compare them with MuCO on CPSea-PDB. The evaluation metrics include success rate, potential energy, and the JS similarities of backbone torsions, side-chain torsions, and secondary structures.
>
> |CPSea-PDB|Success Rate|Energy|Backbone JS Sim.|Side-chain JS Sim.|Secondary Structure JS Sim.|
> |:---|:---:|:---:|:---:|:---:|:---:|
> |**AlphaFold3**|**0.965**|-114.53|0.548|0.464|0.686|
> |**ProteinZen**|0.494|454.76|0.416|0.326|0.700|
> |**MuCO**|0.941|**-160.71**|**0.589**|**0.479**|**0.852**|
>
> The above results show that while AlphaFold3 achieves a success rate comparable to AfCycDesign and HighFold2 (slightly higher than MuCO), it still struggles to accurately recover secondary structures. In single-sampling mode, MuCO achieves the lowest energy profiles and superior JS similarities for torsions and secondary-structures compared to the baselines.
>
> **Q3: How to handle non-canonical amino acids.**
>
> **A3:** For non-canonical amino acids, inspired by HighFold2, some of them (23 types) can be represented and generated as rigid frames with several torsion angles if sufficient training data are available. In Stage 3, the CHARMM36 force field computation supports many non-canonical residues, such as N-methylation.
>
> However, we also acknowledge that it is still challenging to cover all non-canonical amino acids due to data scarcity and residue diversity (non-canonical residues account for 0.4% of residues in the Protein Data Bank, while 78% of these residues are unclassified and the rest comprises 2000+ distinct known types).
>
> **Q4: Does the 3-stage framework suffer from error accumulation?**
>
> **A4:** We acknowledge that generation in Stages 1 and 2 may introduce geometric deviations. That's why Stage 3 (physics-aware optimization) serves as a physical inductive bias to alleviate error accumulation. As shown in Figure 1, Stages 1 and 2 explore the energy landscape to identify promising potential wells, and Stage 3 uses force-field optimization to find valid local minima, effectively offsetting prior errors. As shown in Figure 8 and Table 9, raw outputs are often generally coiled, but the secondary structures are recovered after Stage 3. This phenomenon is not observed in other baselines, such as HighFold2.
>
> **Q5: What if the linear peptide cannot be cyclized stably?**
>
> **A5:** Indeed, certain sequences may be inherently unstable as cyclic peptides. Within the MuCO framework, Stage 3 (Physics-Aware Optimization) acts as a physical validity filter. If a generated conformation is physically unstable, it will exhibit high potential energy or violate geometric constraints following relaxation. As shown in Appendix B.2 and D.2, structures that exceed the predefined energy threshold or bond-length tolerances are formally classified as cyclization failures.
>
> We hope the above replies can resolve your concerns. Thanks in advance for considering raising your score.

---

> > ### Author Rebuttal · Reviewer_A43d · 2026-04-02
> >
> > Thank the authors for the response. My concerns are addressed and I will keep my rating.

---

> > > ### Author Response · Authors · 2026-04-05
> > >
> > > Thank you very much for your time. We appreciate your constructive suggestions which have significantly strengthened the evaluation of our work. And we appreciate your valuable insights and your support for our paper.

---

### Official Review · Reviewer_yYzu · 2026-03-13

**Soundness:** 2
**Presentation:** 3
**Significance:** 2
**Originality:** 3
**Overall Recommendation:** 4
**Confidence:** 2

**Summary:**

MuCO is a generative framework for cyclic peptide cyclization that decouples the task into three sequential stages: SE(3) flow matching-based cyclic backbone generation, torsional flow matching-based side-chain packing with Cyclic RPE, and CHARMM36 force field refinement. By training directly on the CPSea cyclic peptide dataset, the model implicitly learns ring-closure constraints without relying on hard geometric post-processing. A hierarchical K×M parallel sampling strategy enables efficient exploration of the conformational landscape, achieving an amortized inference time of ~41ms per conformation. Experiments on CPSea demonstrate that MuCO significantly outperforms AF2-based and GDL-based baselines in physical stability and structural diversity, while avoiding the mode collapse that renders existing methods incapable of recovering non-H2T cyclization modes.

**Compliance With Llm Reviewing Policy:**

Affirmed.

**Final Justification:**

Most of my concerns have been addressed through the rebuttal.

**Key Questions For Authors:**

The paper's stated motivation centers on drug discovery and therapeutic design of cyclic peptides. However, the evaluation is entirely based on structural metrics (potential energy, Shannon entropy, secondary structure composition). Could the authors provide results on pharmaceutical property metrics such as synthetic accessibility (SA), or membrane permeability for the generated conformations? Without these, it is difficult to assess whether the generated structures are practically useful in a drug discovery context.

**Limitations:**

see weakness.

**Strengths And Weaknesses:**

As shown in Table 10, MuCO's raw output prior to Stage-3 contains ~0% α-helix and ~99% random coil, meaning biologically relevant structure emerges almost entirely through CHARMM36 optimization. The critical missing experiment is a comparison against random initialization followed by CHARMM36 minimization. Without this control, it cannot be determined whether Stages 1 and 2 provide meaningful conformational guidance beyond naive random seeding.

The ablation replacing Stage-2 with EGNN simultaneously removes both flow matching and Cyclic RPE, making it impossible to isolate the contribution of each. A dedicated ablation (flow matching with linear RPE versus flow matching with Cyclic RPE) is necessary to substantiate the claims made in Remark 2.

MuCO generates C2C (disulfide) conformations at 0–1% across all test sets, against a ground truth rate of 2–8%. This failure is not discussed in the main text despite disulfide-bridged cyclic peptides being pharmacologically important. The paper should explicitly address whether this stems from data imbalance in CPSea or from limitations in the topology detection algorithm.

CPSea-PDB, the only experimentally validated subset, contains only 85 samples. Drawing conclusions on secondary structure recovery and cyclization mode coverage from 85 crystallographic structures is statistically fragile for the fine-grained claims made in the paper.

---

> ### Author Rebuttal · Authors · 2026-03-29
>
> Thank you for your appreciation of our work. Below, we answer your questions one by one.
>
> **Q1: Add one more baseline: Random Initial Conformation + CHARMM36.**
>
> **A1:** We compared MuCO against the suggested baseline using CPSea-PDB. We conducted 20 independent trials with random initialization to evaluate changes in the secondary structure (SS) ratio before and after applying CHARMM36 (denoted as $\Delta$SS).
>
> | Condition | SS ratio before applying CHARMM36 | SS ratio after applying CHARMM36 | $\Delta$SS (Gain)
> | :--- | :---: | :---: | :---: |
> | Random (Single) | 0.0314 | 0.0272 | -0.0042
> | Random (Average) | 0.0436 | 0.0194 | -0.0241
> | Random (Best) | 0.0436 | 0.0796 | 0.0360
> | **MuCO** | 0.0080 | 0.2839 | **0.2759**
>
> Here, **Single** means a single random initialization trial using a fixed seed. **Average** means the average gain in secondary structure over trials across 20 seeds. **Best** means the maximum gain in secondary structure observed over these 20 trials.
>
> We can find that 1) random initialization fails to generate secondary structure during relaxation, while MuCO achieves a high $\Delta$SS. 2) CHARMM36 can improve the SS ratio only when good initial conformations are provided --- that is why the Stages 1 and 2 are important.
>
>
> **Q2: The ablation study demonstrating the rationality of cyclic RPE.**
>
> **A2:** We introduced cyclic RPE to capture the local geometric information of cyclic peptides (i.e., the torsions $\chi_1,...,\chi_4$), inspired by AfCycDesign and HighFold2. For the ablation study, we evaluate linear RPE and cyclic RPE on the CPSea-PDB dataset and testify that cyclic RPE slightly outperforms linear RPE on side-chain Jensen-Shannon (JS) similarity (i.e., 1-JS divergence) against the ground truth. Furthermore, different kinds of RPE distinguish linear and cyclic chains.
>
> | Ablation Setting | $\chi$ JS | $\chi_1$-$\chi_2$ JS |
> | :--- | :---: | :---: |
> | MuCO (linear RPE) | 0.8867 | 0.4779 |
> | **MuCO (cyclic RPE)** | **0.8932** | **0.4797** |
>
> Here, $\chi$ JS is the JS similarity of $\chi_1$-$\chi_4$ with ground truth. Because most residues only have two torsions, we further consider $\chi_1$-$\chi_2$ JS, the JS similarity of $\chi_1$-$\chi_2$ joint distribution with ground truth. Notably, when cyclic RPE is adopted in cyclic chains, we can flexibly extend this framework to multi-chain protein-cyclic peptide complexes, which we leave for future work.
>
> **Q3: C2C (disulfide) conformations at 0-1% vs GT 2-8%.**
>
> **A3:** We acknowledge that MuCO yields a limited disulfide conformation, which is important in pharmaceutical practice in the single sampling mode. This stems from the low frequency of disulfide cyclization in datasets, as shown in Appendix C, Figure 11. **Notably, we covered almost all disulfide cases when we employed the hierarchical sampling strategy, as shown in Appendix F, Table 7.**
>
> **Q4: The concern about the limited size of CPSea-PDB.**
>
> **A4:** In our work, CPSea-PDB serves as a gold-standard dataset to test MuCO's generalization ability. **While our main conclusions on secondary structure recovery and mode coverage are primarily drawn from the larger CPTrans and CPBind datasets, as shown in Appendix F, Tables 4-5.** Sorry for the confusion, and we will introduce our experimental settings more clearly in the main paper.
>
> **Q5: Consider evaluating pharmaceutical properties.**
>
> **A5:** We appreciate your suggestion to evaluate pharmaceutical properties.
>
> Firstly, we should note that Synthetic Accessibility (SA) is a sequence-dependent metric that is inapplicable to our work. In particular, MuCO focuses on conformation generation for a fixed sequence, while SA is determined by the input sequence rather than the generated 3D structure.
>
> Secondly, to resolve your concern, we have expanded our evaluation metrics to some key physicochemical properties, including H-bond (H-bond counts in cyclic peptide), $R_g$ (radius of gyration, $Å$), and SASA (solvent-accessible surface area, $Å^2$) on mean deviation and RMSD. These metrics are essential for assessing the structural stability, compactness, and solubility of cyclic peptides, all of which are critical determinants of their metabolic stability and membrane permeability. The results below clearly demonstrate MuCO's superiority. We will add the results in the revised paper.
>
> | CPSea-PDB | H-bond Deviation | H-bond RMSD | $R_g$ Deviation | $R_g$ RMSD | SASA Deviation | SASA RMSD |
> | :--- | :---: | :---: | :---: | :---: | :---: | :---: |
> | **EGNN** | 2.8228 | 4.3239 | 0.5395 | 0.7441 | 144.0284 | 189.6571 |
> | **WGFormer** | 3.1690 | 4.0704 | 0.6680 | 0.7192 | 178.2712 | 190.8699 |
> | **AfCycDesign** | 3.9753 | 4.9383 | 0.5621 | 0.8723 | 266.6482 | 302.3426 |
> | **HighFold2** | 4.6471 | 4.8353 | 0.5936 | 0.7798 | 272.1592 | 279.9198 |
> | **MuCO** | **1.4250** | **3.6030** | **0.2810** | **0.6479** | **93.6179** | **123.0153** |
>
> We hope the above responses help enhance your confidence to further support our work.

---

> > ### Author Rebuttal · Reviewer_yYzu · 2026-04-05
> >
> > Thank the authors for the response. My concerns are addressed and I will keep my rating.

---

> > > ### Author Response · Authors · 2026-04-05
> > >
> > > Thank you very much for your time. We appreciate your constructive suggestions which have significantly strengthened the evaluation of our work. And we appreciate your valuable insights and your support for our paper.

---

### Official Review · Reviewer_Lv6n · 2026-03-13

**Soundness:** 3
**Presentation:** 3
**Significance:** 3
**Originality:** 3
**Overall Recommendation:** 5
**Confidence:** 3

**Summary:**

The work presents MuCo, a method to design cyclic peptides through multiple stages involving different constrain.

**Compliance With Llm Reviewing Policy:**

Affirmed.

**Final Justification:**

The author addressed my questions well. The design pipeline is under well consideration.

**Key Questions For Authors:**

1. MuCO samples $K$ backbone structures and $M$ side-chain conformations per backbone. How should these parameters be chosen in practice to balance generation quality and computational cost?
2. Ring closure is claimed to emerge implicitly from training on cyclic data, yet there exists 5%-6% failure rate. How do authors guarantee ring-closed outputs without an explicit geometric loss term?
3. MuCO conditions on a linear peptide. How strongly does the initial linear precursor constrain the generated cyclic ensemble? Is it only providing sequence length context, or does it really restrict the accessible cyclic structures?
4. Stage 2 constructs the cyclic residue graph treating residue 1 and $L$ as neighbors, which corresponds to a head-to-tail cyclication. However, for Cys-cys or isopeptide modes, the ring connects internal residues. How is the Stage 2 cyclic graph constructed when the cyclization mode is unknown at inference time, given that Stage 3 appears to detect the closure mode only afterward?
5. For the training data, how are "high-confidence" pairs defined?

**Limitations:**

Since MuCO relies on ESM2 embeddings as sequence representations, the method may primarily explore conformations for the natural amino acids rather than performing fully unconstrained sequence design. Clarifying the extent to which the method can explore novel sequence space would strengthen the paper.

**Strengths And Weaknesses:**

**Soundness**
The overall algorithmic design is reasonable, and multi-stage optimization procesure is logically motivated.

**Presentation** Consistent and easy to follow.

**Significance** Important for cyclic peptide design, which has therapeutic potential regarding protease resistance.

**Originality**
This work proposes a multi-stage refinement pipeline for cyclic peptide generation.

**Strength** The idea of swapping the linear positional attention graph with the cyclic one is a notable engineering choice.

**Weakness** See questions

---

> ### Author Rebuttal · Authors · 2026-03-29
>
> Thank you for your appreciation of our work. Please find our detailed responses to your specific concerns below.
>
> **Q1: How to select $K$ and $M$?**
>
> **A1:** As presented in Figure 6, Figure 7, and Table 7 in our paper, MuCO samples achieve significantly lower energy and 100% cyclization success when $K=5$ and $M=5$. In practice, $K=3$ and $M=3$ are sufficient to achieve low energy, 100% cyclization success, and recognition of diverse cyclization modes in most cases. We should note that disulfides are more challenging to detect—this stems from the low frequency of disulfide cyclization in the datasets (Appendix C), so we recommend $K=5$ and $M=3$ when Cys-Cys pairs occur. In this setting, the sampled disulfide distribution approximates the ground truth distribution without high computational cost during the relaxation phase. We will add the above content to the revised appendix.
>
> **Q2: How to guarantee achieving ring-closed outputs?**
>
> **A2:** As shown in Appendix A (Eq.(5)), the loss function of backbone flow matching is
>
> $\mathcal{L}\_{backbone}=\mathcal{L}\_{trans}+0.5\mathcal{L}\_{rot}+\mathcal{L}\_{bbatom}+\mathcal{L}\_{distmat}+0.25\mathcal{L}\_{aux}$.
>
> Here, the distance matrix loss $\mathcal{L}_{distmat}$ explicitly designates pairwise distance constraints between terminal residues and geometrically, which guarantees this ring closure.
>
> **Q3: Does the linear precursor restrict structures?**
>
> **A3:** In MuCO, the linear precursor primarily provides sequence-level context by ESM2 embeddings rather than restricting the final cyclic structures. However, for baselines such as EGNN and WGFormer, the linear precursor provides conformational information to initialize message passing.
>
> **Q4: In Stage 2, cyclic graph construction is achieved without prior knowledge of cyclization mode.**
>
> **A4:** Our current dataset contains only head-to-tail cyclization with different modes, which is also a more accessible and widely adopted cyclization strategy in wet lab experiments. As a result, the cyclic RPE we used is primarily defined for head-to-tail connections.
>
> Notably, for other cyclization modes (e.g., internal connections), cyclic RPE is also applicable, which has been demonstrated by HighFold2. In other words, cyclic RPE enables flexible extensions to more diverse cyclization when corresponding training data becomes available.
>
> **Q5: How to obtain "high-confidence" pairs in training data?**
>
> **A5:** As shown in Appendix C, we select peptides with lengths $L \in [8, 16]$, a range that is most important for pharmaceutical applications. Then we filter cyclic peptides whose relative molecular quantities are less than 0.2. As shown in Section 4.1, we then generate conformations for corresponding linear peptides using SimpleFold 3B.
>
> Regarding MuCO's limitation, we would like to clarify that MuCO is primarily designed for exploring conformation landscapes rather than unconstrained sequence design. For non-canonical amino acids, inspired by HighFold2, some of them (23 types) can be represented and generated as rigid frames with several torsion angles when corresponding training data are available. In Stage 3, the CHARMM36 force field computation supports many non-canonical residues, including N-methylation.
>
> We acknowledge that it is still challenging to cover all non-canonical amino acids due to data scarcity and residue diversity (non-canonical residues account for 0.4% of residues in the Protein Data Bank (PDB), of which 78% are unclassified, and the rest comprises 2000+ distinct known types). We will update the limitation section to highlight this clarification.
>
> We hope the above replies can resolve your concerns and make you more confident in supporting our work. Thank you again for your appreciation and constructive suggestions.

---

> > ### Author Rebuttal · Reviewer_Lv6n · 2026-04-03
> >
> > The author addressed my questions and also the limitations. I will raise my score.

---

> > > ### Author Response · Authors · 2026-04-05
> > >
> > > Thank you very much for your constructive feedback and willingness to raise the score. We are glad that our reply has successfully addressed your concerns. We will ensure that all the discussed details are carefully incorporated into later versions.

---

### Decision · Program_Chairs · 2026-04-30

**Decision:**

Accept (regular)

**Comment:**

This paper presents a three-stage pipeline for cyclic peptide conformation sampling (backbone generation --> side-chain packing --> force field relaxation), targeting a therapeutically relevant and under-explored problem. The manuscript is in general well-written and easy to follow, and the model showed strong empirical performance on a clinically meaningful problem.

**Rebuttal summary** Reviewers' concerns were centered on metric selection, generalization to non-standard amino acids, hallucinations on non-cyclizable cases, and ablations decoupling model-vs-random initialization and cyclic RPE effects. Authors adequately addressed these with additional experiments and clarifications; meanwhile scope limitations (single-chain, head-to-tail cyclization, data constraints) are appropriately acknowledged (should update in the revision). Heavy reliance on Stage 3 for structural refinement and hallucination filtering is a limitation, though acceptable given the current scope.

**Recommendation: Accept** The pipeline is competent despite using existing methods; the problem is well-motivated, and sufficiently supported empirically. Remaining limitation do not undermine the core contributions.